

# Testing data assimilation strategies to enhance short-range AI-based discharge forecasts

Bob E. Saint-Fleur[1], Eric Gaume[1], Florian Surmont[1], Nicolas Akil[2], and Dominique Theriez[2]

[1]GERS-EE, Université Gustave Eiffel, Allée des Ponts et Chaussées, 44344 Bouguenais, France
[2]Aquasys Entreprise, 2 rue de Nantes, 44710 Port-Saint-Père, France

**Correspondence:** Bob E. Saint-Fleur (bob.saint-fleur@univ-eiffel.fr)

**Abstract.** Effective discharge forecasts are essential in operational hydrology. The accuracy of such forecasts, particularly in short lead times, is generally increased through the integration of recent measured discharges using data assimilation (DA) procedures. Recent studies have demonstrated the effectiveness of deep learning (DL) approaches for rainfall-runoff (RR) modeling, particularly Long Short-Term Memory (LSTM) networks, outperforming traditional approaches. However, most

of these studies do not include DA procedures, which may limit their operational forecast performance. This study suggests and evaluates three DA strategies that incorporate discharge from either past observed discharges or forecast discharges of a pre-trained benchmark model (BM). The proposed strategies, based on a Multilayer Perceptron (MLP) orchestrator, include: (1) the integration of recent observed discharges, (2) the integration of both recent discharge observations and pre-trained BM forecasts, and (3) the post-processing of BM forecast errors. Experiments are implemented using the CAMELS-US dataset

using two established benchmark models: the trained LSTM model from Kratzert et al. (2019) and the conceptual Sacramento Soil Moisture Accounting (SAC-SMA) model from Newman et al. (2017), covering both machine learning and conceptual RR simulation approaches. Lead times of 1, 3, and 7 days, covering short- and mid-term horizons, are considered. The approaches are evaluated in two forecast frameworks: (1) perfect meteorological forecasts over the forecasting lead time and (2) highly uncertain ensemble meteorological forecasts. The two frameworks yield contrasting outcomes. When evaluated under the

perfect forecast framework, the application of DA leads to substantial improvements in forecast performance, although the magnitude of these gains depends on the initial performance of the benchmark (BM) models and the forecasting lead time. Improvements are consistently significant for the SAC-SMA cases, while for the LSTM cases, gains are observed mainly for basins where the LSTM initially underperforms. However, the ensemble forecast evaluation yields unexpected results: the performance ranking of the tested models changes markedly compared to the perfect forecast framework. The LSTM model,

in particular, appears penalized by the unreliability - specifically, the under-dispersion - of its forecast ensembles, meaning that its predictions are insufficiently responsive to meteorological forcing over the forecast lead time. This finding underscores the importance of ensuring reliable ensemble dispersion for the efficient operational deployment of AI-based hydrological forecasts.



## 1 Introduction

Discharge forecasting models are essential in operational hydrology, whether for water resource or related-risk management. Their importance is set to increase as climate-related threats intensify (Schiermeier, 2018; Philip et al., 2020; Rentschler et al., 2023). However, providing accurate discharge forecasts remains challenging due to the complexity of rainfall-runoff (RR) processes, model imperfections, and uncertainty in input data, particularly in weather forecast quality.

Over the years, significant efforts have been made to address the challenges of hydrological modeling, leading to the development of various models and approaches. In the era of artificial intelligence (AI), notable advances have been achieved, with recent studies demonstrating the outstanding performance of deep learning models (DL) relative to traditional RR models (Kratzert et al., 2019; Husic et al., 2022). Commonly used DL architectures include multilayer perceptrons (MLPs) (Jeannin et al., 2021; Saint-Fleur et al., 2023), recurrent neural networks (RNNs) such as Long Short-Term Memory (LSTM) networks

(Kratzert et al., 2018, 2019; Fang et al., 2021; Wunsch et al., 2021; Rahbar et al., 2022), and more recently, Transformers (Pölz et al., 2024).

Despite these advances, most hydrological models in the literature mainly focus on discharge simulation rather than forecasting, which is a fundamentally different task. Discharge simulation involves replicating a hydrosystem's behavior using observed meteorological input, while forecasting aims to predict future discharge values at specific lead times, often relying

on inputs subject to considerable uncertainty. Although simulation models can be integrated into forecasting systems, either as assimilable data or as driven by forecasted forcings, their development frequently overlooks key components such as data assimilation (DA), persistence analysis, and ensemble (probabilistic) assessment.

Persistence analysis, introduced by Kitanidis and Bras (1980), evaluates a model's performance relative to a naive baseline, which simply translates the current observation to the target lead time. This analysis, which serves as a relevant benchmark for

assessing the predictive ability of models, is rarely considered in most hydrological modeling studies.

Data assimilation (DA), on the other hand, which consists of dynamically providing real-time observations to a running forecast model, is essential in operational forecasting (Bourgin et al., 2014; Boucher et al., 2020; Piazzi et al., 2021). By ensuring regular updates of the model states, DA allows one to reduce the impact of uncertainties associated with meteorological forecasts and model structures, thus keeping the model aligned with evolving hydrological conditions. Several DA techniques

exist, and their efficacy often depends on the reliability of the underlying model. For direct discharge assimilation strategies, the importance of DA is typically more pronounced at shorter lead times. However, suboptimal models may over-rely on the assimilated discharge data, which may shadow the contribution of the forcings, leading toward naive models (Saint Fleur et al., 2020). Thus, DA methods can improve the operational application of RR forecasting models but are not straightforward to calibrate and implement efficiently.

In the following, two benchmark models are considered to evaluate the added value of DA procedures: the regional LSTM model of Kratzert et al. (2019) and the basin-specific conceptual SAC-SMA model from Newman et al. (2017). A preliminary analysis of the performance of both models when implemented with the CAMELS-US data set (Newman et al., 2014, 2022; Addor et al., 2017) is presented in Fig. 1. Using a perfect weather forecast to provide a one-day-ahead discharge forecast (see





Sect. 2.3 for implementation details), we calculate the classical Nash-Sutcliffe efficiency (NSE) (Nash and Sutcliffe, 1970) and

the Persistence Criterion (PERS). The results indicate that $NSE \geq 0.6$ is achieved in 90 % and 40 % of the tested basins for the LSTM and SAC-SMA models, respectively. However, when using the PERS criterion, the proportion of basins meeting this level drops to 60 % for LSTM and 15 % for SAC-SMA. Furthermore, a $PERS \leq 0$ is observed, indicating that the mean squared error of the model exceeds that of the naive model, occurring in 20 % of the basins for LSTM and 40 % for SAC-SMA. These observations suggest that, at least for these basins, DA methods could improve the quality of operational forecasts

generated by either of these RR models.

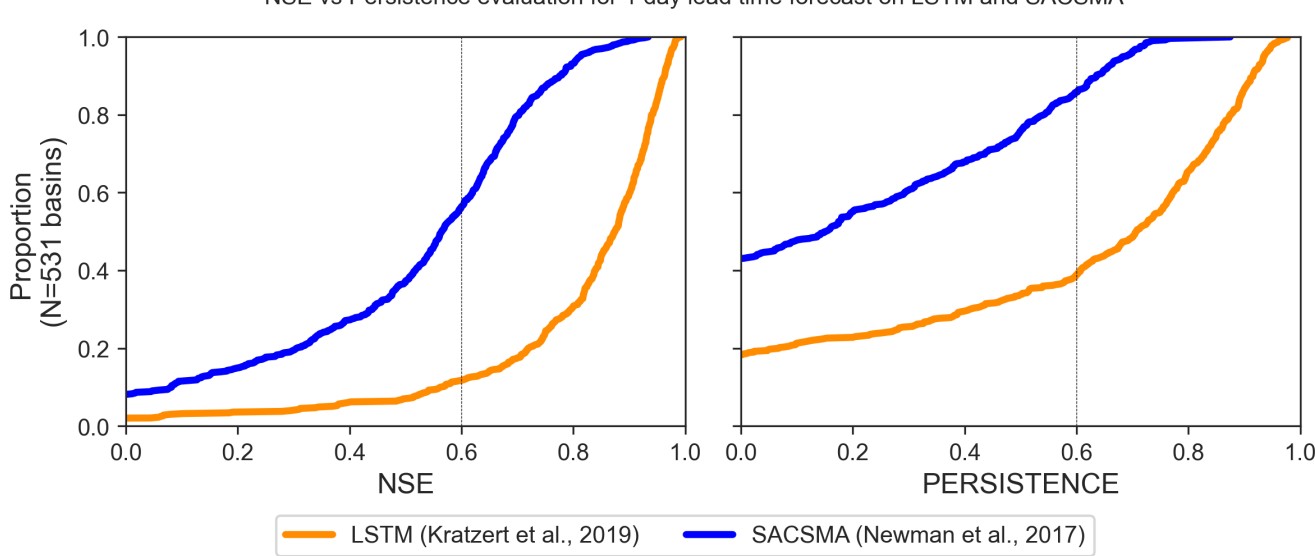

**Figure 1.** NSE and Persistence analysis on Benchmark models

Three different data assimilation (DA) strategies, that take into account past observed discharges to generate forecasts, will be tested. For simpler implementation, including time and resource efficiency, a MultiLayer Perceptron (MLP) is used as the orchestrator in these DA methods.

As assimilated discharge procedures generally lose effectiveness at extended lead times, forecasts are considered at both

short- and mid-term lead times. These lead times are defined with respect to the basin response times estimated on the basis of a rainfall-discharge cross-correlation analysis. To ensure operational relevance and reflect real-world forecasting practices, two scenarios are considered with respect to weather forecasts: (1) weather forecasts will be assumed to be perfect, (2) weather forecasts will be assumed to be highly uncertain, and ensemble forecasts will be considered. Forecast performance is evaluated using both deterministic and probabilistic criteria.

This paper is structured as follows : Section 2 introduces the data and the benchmark models. Section 2.2 presents the data set, the proposed data assimilation methods, the experimental forecasting setup, and the evaluation metrics. The results



for the deterministic and ensemble forecasts are successively presented and discussed in Sect. 3. Section 4 presents the main conclusions.

## 2 Materials and Methods

### 2.1 Dataset


The CAMELS-US dataset (Newman et al., 2014, 2022; Addor et al., 2017) consists of basin-averaged hydrometeorological time series, catchment attributes, and daily streamflow observations from the United States Geological Survey (USGS) for 671 catchments across the Contiguous United States (CONUS). The meteorological forcings are available from either Daymet, NLDAS and Maurer sources. As this study stands on the benchmark work of Kratzert et al. (2019) and Newman et al. (2017) - hereafter tracked as LSTM and SAC-SMA - it is limited to the same subset of 531 basins, the Maurer forcings, and the 1989-2008 period used in these previous works. The experiments developed hereafter use the 1989-2006 period as the training subset, and the remaining 2006-2008 as the test subset. Using pre-trained benchmark models (SAC-SMA from Newman et al. (2017) and the LSTM from Kratzert et al. (2019)), two distinct time series of discharge have been re-simulated on the whole 1989-2008 period and used to complement the dataset, see Table 1.


**Table 1.** Available time series for the 531 basins over the period 1989-2008

| Type | Variables | Description | Unit | Source |
|---|---|---|---|---|
| Forcings | PET | Potential Evapotranspiration | mm/day | Maurer |
| | PRCP | Rainfall | mm/day | " |
| | SRAD | Incident Solar radiation | $W/m^2$ | " |
| | Tmax | Daily maximum temperature | °C | " |
| | Tmin | Daily minimum temperature | °C | " |
| | Vp | Vapor Pressure | Pa | " |
| Target variable | Q.OBS | Observed discharge | mm/day | USGS |
| Model outputs | Q.SAC | SAC-SMA simulated discharge | mm/day | Newman et al. (2017) |
| | Q.LSTM | LSTM simulated discharge | mm/day | Kratzert et al. (2019) |


The added value of the proposed data assimilation strategies will be evaluated for two types of RR models: (a) the LSTM proposed in Kratzert et al. (2019), which was trained regionally and incorporates static basin-specific inputs, and (b) the conceptual global model SAC-SMA from Newman et al. (2017). As in Kratzert et al. (2019), the SAC-SMA model has been chosen as a reference to illustrate the performance of conceptual RR models, which remain widely used for operational discharge forecasting.





## 2.2 Data assimilation procedures


To take away any confusion, the term "data assimilation" used in this study is based solely on the integration of the recent discharge data. Therefore, it could also be termed "discharge assimilation". In that sense, three data assimilation procedures are tested, integrating either recent discharge measures or simulations from the two RR models. **Multilayer perceptron (MLP)** networks (Rosenblatt, 1958) are used as orchestrators in these three strategies; this choice is primarily motivated by computa-

tional efficiency. MLPs have become largely adopted over recent decades (Werbos, 1988a, b, 1974), and several studies have shown their effectiveness in RR modeling (Atmaja and Akagi, 2020; Oliveira et al., 2021; Jeannin et al., 2021; Saint-Fleur et al., 2023). Although recent studies have demonstrated the superior performance of models such as LSTM (Kratzert et al., 2018) networks or transformers (Li et al., 2024), MLPs have been used in this study not only as a forecast orchestrator, but also as a RR modeling alternative due to the relative simplicity of their implementation.

The three data assimilation procedures are summarized in Fig.2 and are described straight after:

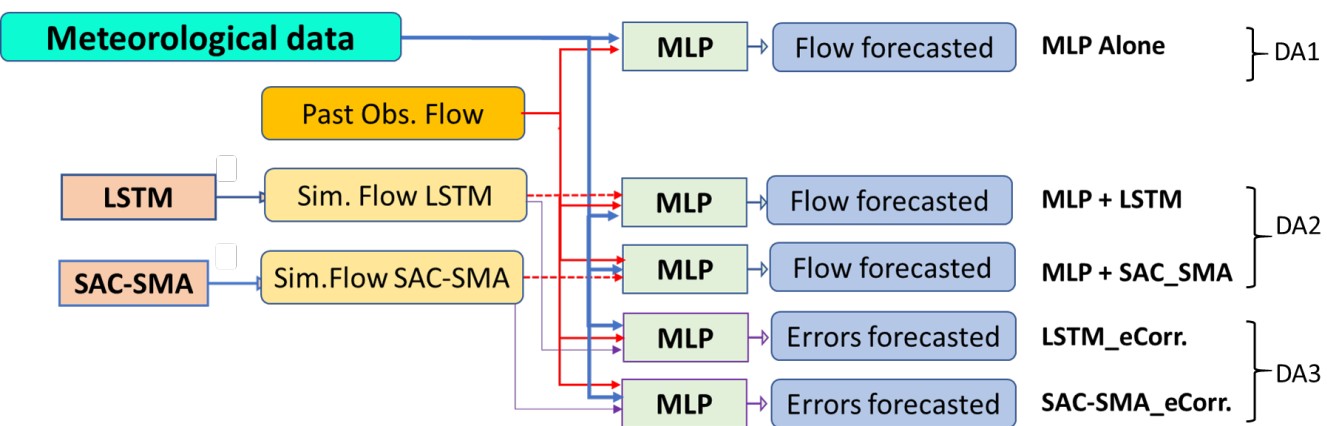

**Figure 2.** Data assimilation set-up: DA1, MLP Alone; DA2,MLP fed with RR model forecasts (MLP+LSTM or MLP+SAC-SMA); DA3, Post-treatment of RR forecasting errors noted as LSTM_eCorr and SAC-SMA_eCorr.


1. DA-1: Direct forecast of discharges $\hat{Q}_{t+hp}$ over the forecast horizon $hp$ with an MLP, fed with the past observed discharges $Q^o$, observed meteorological variables $X^o$, as well as meteorological forecasts $\hat{X}$ (see Eq.1).

$$\hat{Q}_{t+hp} = f(Q^o_{t-p:t}, \hat{X}_{t-n:t+hp}, X^o_{t-n:t}) \tag{1}$$

2. DA-2: The same approach as in DA-1 but with the forecasts of the RR model $Q^s$ (either SAC-SMA or LSTM) as

additional input variables (see Eq.2).

$$\hat{Q}_{t+hp} = f(Q^s_{t-p:t+hp}, Q^o_{t-p:t}, \hat{X}_{t-n:t+hp}, X^o_{t:t-n}) \tag{2}$$

3. DA-3: Post-processing of the prediction errors of the RR model $\varepsilon_t$ (again SAC-SMA or LSTM). In this strategy, the orchestrator is used to forecast the errors ($\hat{\varepsilon}_{t+hp}$) of the RR model over the horizon $hp$ and the prediction errors are then





added to the forecasts of the RR model. The assimilation procedure then proceeds in three steps (see Eq.3, Eq.4, and
Eq.5).

$$\varepsilon_t = Q_t^o - Q_t^s \tag{3}$$

$$\hat{\varepsilon}_{t+hp} = f(\varepsilon_{t-p:t}, Q_{t-p:t}^o, \hat{X}_{t-n:t+hp}, X_{t-n:t}^o) \tag{4}$$

$$\hat{Q}_{t+hp} = Q_{t+hp}^s + \hat{\varepsilon}_{t+hp} \tag{5}$$

In the previous equations, $n$ and $p$ are the sequence lengths for the forcing and the assimilated discharge. These values
will be fixed based on the mean response time of the basins using a RR cross-correlation analysis, see Fig.5. As suggested in
Saint Fleur et al. (2020), to prevent the models from relying disproportionately on assimilated discharge rather than forcing,
we imposed $n \geq p$.

In summary, seven (7) different model configurations are compared: the five (5) data assimilation procedures (unfolded
from DA1, DA2, DA3) presented in this section, plus the two (2) direct forecasts from both pre-trained models, SAC-SMA and
LSTM, which serve as benchmarks to evaluate the efficiency of the tested data assimilation strategies. The direct forecasts from
the benchmark models were assumed to be unchanged for the tested lead time; therefore, no further running was necessary.

In both forecasting approaches and for each basin, the MLPs were trained (i.e., calibrated) 60 times with a random selection
of their initial parameter values (seeds), leading to 60 different possible trained models. Likewise, 8 seeds have been considered
for the LSTM and 10 for the SAC-SMA model. This aims to account for the uncertainties and variability induced by model
initialization during training. The assimilation strategies are trained on the basis of the series of median simulated values of
both models (SAC-SMA and LSTM). The predictions thus consist of an ensemble of 60 runs for the assimilation strategies
and, respectively, 8 and 10 runs for the LSTM and SAC-SMA benchmark forecasts without assimilation. The performances of
the ensemble simulations (dispersed by random initialization) are analyzed, based on their median values, in the first part of
this paper (Sect. 3.1) and in figure 1. In the case of the climatological ensembles, all members of the ensembles are considered
and analyzed in the rest of the paper.

## 2.3 Forecasting setup

As illustrated by the equations Eq.1 to Eq.5, the choice of the input features for a forecasting model that includes data assimi-
lation may be affected by the forecasting lead time $hp$. Hence, a specific model should be calibrated for each considered lead
time. The alternative, consisting of iteratively calibrating the one-step-ahead model toward larger lead times, is inefficient as it
drastically increases the forecast uncertainty (Chevillon, 2007; Teräsvirta et al., 2010; Liu and Wang, 2024). This finding has
been confirmed during the present work (results not presented herein).




Three lead times are explored: 1, 3 and 7 days. In the absence of operational weather forecast archives for the evaluation
period and the considered basins, two forecasting strategies are tested, as illustrated in Fig. 3. First, forecasted weather variables
are considered equal to the actual (upcoming) observed ones at the lead date of the current year. This configuration is hereafter
referred to as the **perfect** (i.e. ideal) weather forecast. Second, ensembles of weather forecasts are resampled from historical
weather records. Various sampling strategies could be considered for this approach, for example, based on similarities between
current and historical hydrological states, regardless of date or season (Hidalgo and Jougla, 2018). But in this study, a simple
strategy has been adopted as forecast members are selected on a date-to-date basis: from a given date ($t_0$) in the evaluation
period (2006-2008), spanned to the lead time ($t_0 + h_p$), a sequence ($t_0 : t_0 + h_p$) is established; the same sequence index (day
and month) is picked at every year in the training/calibration period (1989-2006), then becomes a member of the ensemble
weather forecast (totalized to 18 members). This approach is termed herein as **climatological ensembles**, and will be used for
probabilistic analysis of the ensemble forecasts.

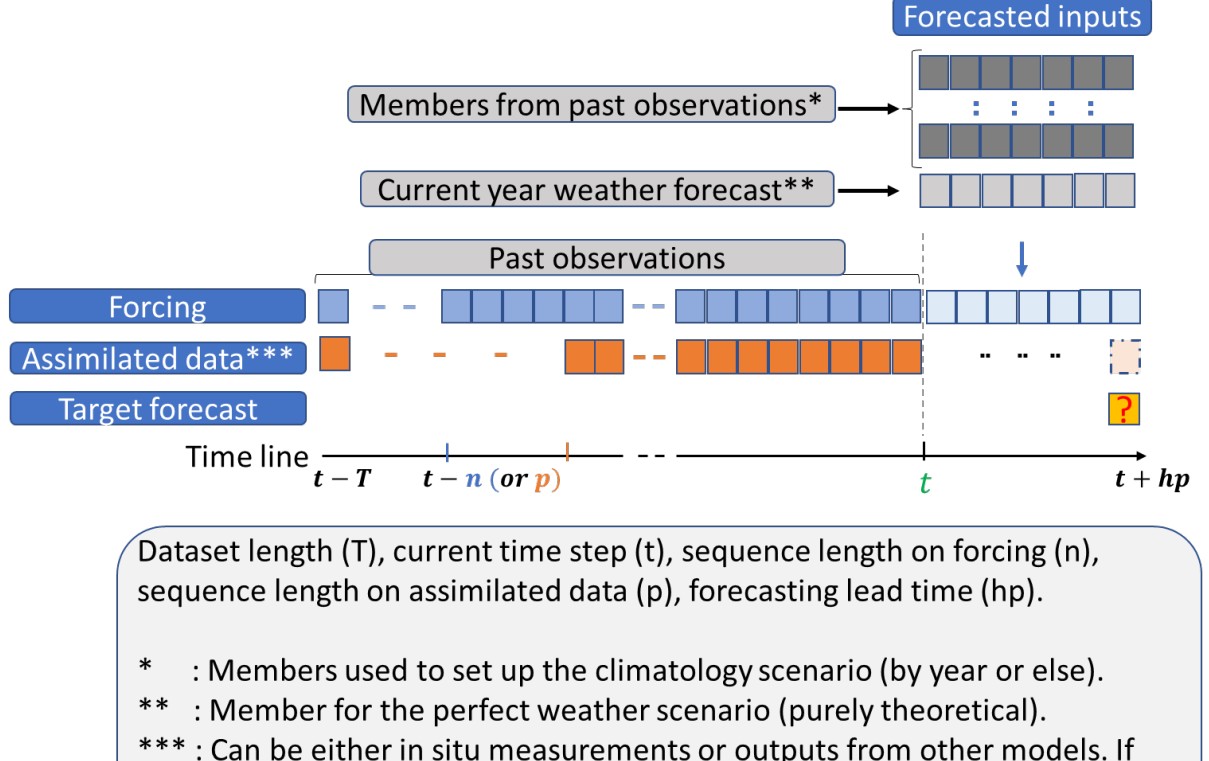

**Figure 3.** Forecasting assumptions setup

All the proposed data assimilation strategies were trained based on the **perfect** weather forecast configuration, and subse-
quently evaluated under both the **perfect** and the **climatological ensembles** configurations.




The implementation in the DA-1 and DA-2 procedures is relatively straightforward in both forecast strategies (perfect or ensemble). For DA-3 under the climatological ensemble scenario, the past error vector $\varepsilon_{t-p:t}$ (in Eq.4) used to adjust a model is that of the perfect scenario.

## 2.4 Evaluation metrics

Numerous metrics are proposed in the literature to evaluate the skills of hydrometeorological forecasting models (Murphy, 1993; Seillier-Moiseiwitsch and Dawid, 1993; Bradley and Schwartz, 2011; Lai et al., 2011; Harold et al., 2015; Petropoulos et al., 2022): evaluating the efficiency for deterministic and ensemble predictions, but also reliability and resolution for ensemble predictions (Bradley and Schwartz, 2011; Slater et al., 2019). The selected evaluation metrics are presented below.

### 2.4.1 Forecasting efficiency

The **efficiency** is a measure of the proximity between the observed values $Q_t$ and the predicted values $\hat{Q}_t$. The commonly used metrics for deterministic forecasts, are based on the sum of square errors: Nash-Sutcliffe Efficiency (NSE), Eq.6 (Nash and Sutcliffe, 1970), the Kling-Gupta Efficiency (KGE) (Gupta et al., 2009) and the Persistency Criterion (PERS), Eq.7 (Kitanidis and Bras, 1980; Corradini et al., 1986; Anctil et al., 2004).

$$NSE = 1 - \frac{\sum_{t=1}^{T}\left(Q_t - \hat{Q}_t\right)^2}{\sum_{t=1}^{T}\left(Q_t - \overline{Q}\right)^2} \tag{6}$$

$$PERS = 1 - \frac{\sum_{t=hp}^{T}\left(Q_t - \hat{Q}_t\right)^2}{\sum_{t=hp}^{T}\left(Q_t - Q_{t-hp}\right)^2} \tag{7}$$

$NSE$ and $PERS$ are scores that measure the proportion of the sum of square errors of an unskilled model explained by the calibrated (or trained) forecasting model. The unskilled benchmark model for NSE is the trivial mean model ($\hat{Q}_{t+hp} = \overline{Q}$), and for PERS the persistent model ($\hat{Q}_{t+hp} = Q_t$). Both criteria range from 1 (perfect model) to $-\infty$. A negative value indicates that the model produces higher errors and, consequently, worse than the unskilled benchmark models. It should be noted that it is more difficult to achieve a positive PERS than a positive NSE, particularly at short lead times.

For ensemble forecasts, the Continuous Ranked Probability Score (CRPS), Eq.8 (Hersbach, 2000; Matheson and Winkler, 1976; Brown, 1974), is commonly used.

$$CRPS = \frac{1}{T}\sum_{t=1}^{T}CRPS_t \quad with \quad CRPS_t = \int_{-\infty}^{\infty}\left[F_t(y) - \mathbf{1}_{\{y \geq Q_t\}}\right]^2 dy \tag{8}$$

Where, for time step $t$, $F_t$ is the cumulative distribution of the ensemble forecasts, $Q_t$ the observed value, $\hat{Q}_t$ the predicted value, $\overline{Q}$ is the time average of the observed values and $\mathbf{1}_{\{y \geq Q_t\}}$ the Heaviside-step function for binary 0|1 outcome. The




CRPS ranges from 0 (perfect models) to $\infty$ (low-quality models). Note that the CRPS is the mean absolute error of the model in the case of a deterministic forecast (i.e. ensemble composed of a unique member).

### 2.4.2 Forecasting reliability


An ensemble forecast is considered reliable if the ensemble spread reflects the range of prediction errors. If so, the position of the observed value in the ensemble (i.e. its rank throughout the members in a sorted disposition) will be uniformly distributed. The resulting distribution of the ranks of a sufficient number of observations, as proposed in (Hamill, 2001; Talagrand et al., 1997), provides a visual verification of the reliability of the ensemble forecasts. The lack of reliability may take different forms:

(i) a tendency to overestimate (resp. underestimate) leading to an over-representation of the lower (resp. higher) ranks in the rank diagram; (ii) under- or over-dispersions of the ensembles, resulting in a *U-shape* or *Dome-shape* of the rank diagrams. Figure 4 shows the rank diagrams of the evaluation period (2006-2008) throughout the remaining period (1989-2006), for the daily rainfall and PET data.

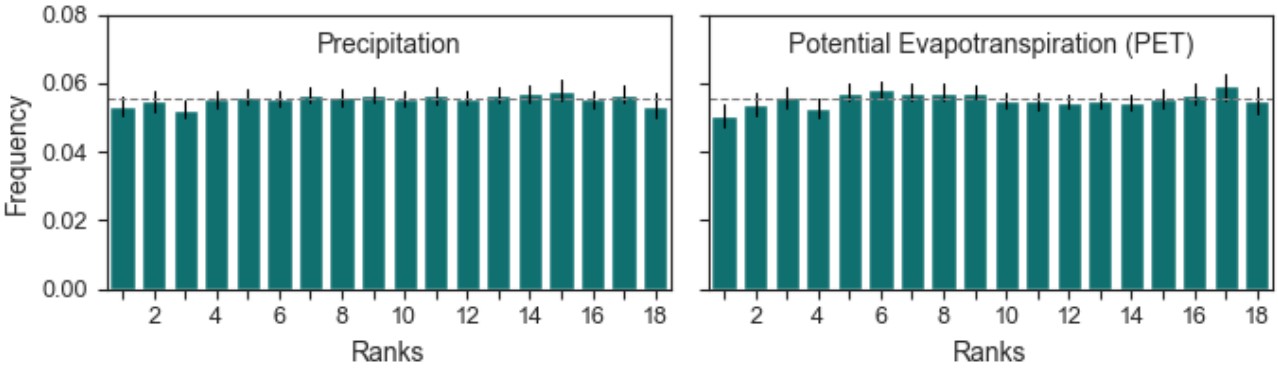

**Figure 4.** Rank diagrams for the daily precipitation and PET climatological ensembles drawn from the period 1989-2005 and evaluated for the test period 2006-2008. The error-bars represent the inter-basin variability, the dashed line shows the theoretical uniform distribution

The rank diagram of the climatological ensembles does not reveal any major deviation from the uniform reference model

(Fig. 4) and hence no obvious biases of the considered ensemble. However, as mentioned by Hamill (2001), global rank diagrams may mask some defaults of the ensembles; therefore, it will be complemented by spread/skill scores.

The spread-skill ratio is a widely used metric to evaluate the reliability of ensemble forecasts. It compares the ensemble spread (the forecast uncertainty), with the actual forecast error (skill) of the ensemble mean. As formalized by Whitaker and Loughe (1998), it is typically calculated as the ratio of the root of the mean of the ensemble variance (spread) to the root mean

squared error (RMSE) of the ensemble mean. Values close to one indicate a well-calibrated ensemble, while values below (above) one reveal under- (over-) dispersion.





### 2.4.3 Forecasting resolution

In ensemble forecast verification, resolution refers to the ability of a model to discriminate events and non-events: i.e. the exceedance or non-exceedance of a given threshold discharge for hydrological predictions. Commonly used metrics for such

evaluation include the Brier score (Brier, 1950) and the AUC score (Area Under the Curve) estimated based on a ROC (Receiver Operating Characteristic) curve.

    – **Brier score (BS)**

$$BS = \frac{1}{N} \sum_{i=1}^{N} (f_i - o_i)^2 \tag{9}$$

$N$ is the number of time steps, $f_i$ is the forecast probability of the event according to the ensemble, and $o_i$ the observed boolean

outcome (1 if the event occurs and 0 otherwise).

    The Brier score values range from 0 (perfect) to 1, and is equal to 0.25 for a random detection model (i.e. the no-skill model).

    – **ROC curves and AUC**

To elaborate a ROC curve, given a selected target discharge threshold, each rank of the ensemble is selected in turn as the forecast value for the event detection. The True positive rate (TPR: proportion of observed events predicted as events) and

the False positive rate (FPR: proportion of non-events predicted as events) are computed for each ensemble rank, over the evaluation period. The ROC curve relates TPR and FPR. The AUC is the estimated area under the ROC curve. It takes its value between 1 (perfect model, TPR=1 and FPR=0 for all ranks) and 0. The ROC curve of a random detection model corresponds to the diagonal (i.e. TPR=FPR=proportion of predicted events). The AUC value of this random detection model is equal to 0.5.

    The resolution measure depends on the chosen discharge threshold. To evaluate the prediction models, several threshold

values will be tested, corresponding to discharge quantiles of the observed series with non-exceedance probabilities $P$ of 0.01, 0.05, 0.10, 0.25, 0.50, 0.75, 0.90, 0.95, and 0.99. For thresholds below the median ($P \leq 0.5$) values, an observed discharge is considered as event, representing low-flow conditions. For higher thresholds ($P > 0.5$), an event is defined as any discharge values exceeding the threshold, corresponding to flood conditions.

## 2.5 Experimental settings

### 2.5.1 Input sequence size and lead time selection strategy

The sizes of the input sequences of the MLPs have been set based on cross-correlation diagrams as suggested by Saint Fleur et al. (2020): see Fig. 5. The median cross-correlation coefficients were considered in the 531 basins. Following Mangin (1984), a limit value has been chosen for the autocorrelation coefficient for discharges of $0.2$ to fix the length $p$ of the input sequence of past observed discharges. Concerning the sequence on the forcing, a size $n$ of 30 past days has been selected as an arbitrary

value. along the flatten ending part of the RR cross-correlogram.





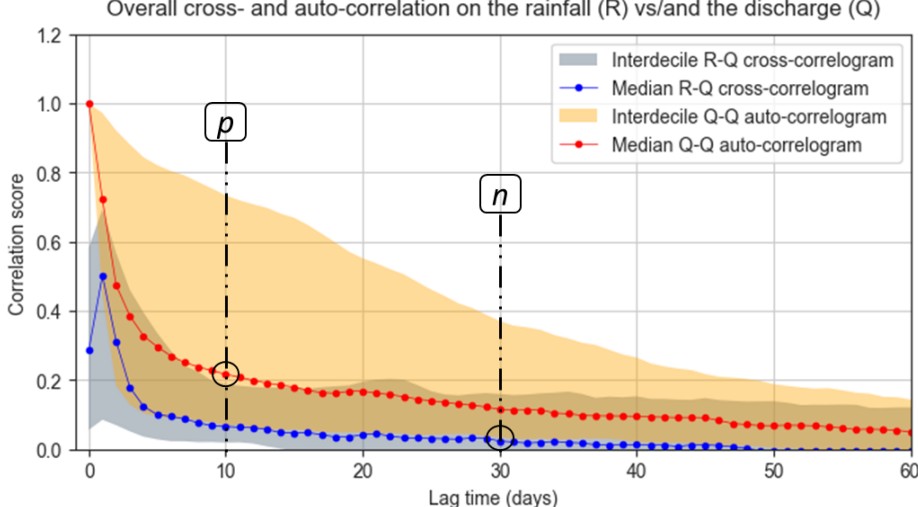

**Figure 5.** Rainfall - Discharge cross-correlation. The chosen sizes ($n$ and $p$) of the input sequences are marked with the dashed-dotted lines.

The correlation coefficients between observed discharges and daily rainfall amounts are the highest for lag times between 1 and 3 days, suggesting that the basins of the CAMELS-US sample have on average short response times, typically less than 3 days. This led us to select three forecast lead times: 1, 3, and 7 days. According to the response times of the basins, it is foreseen that short-term predictions 1 day ahead will be partly controlled by past observed rainfalls, whereas mid-term 3 to
7-day forecasts will be mostly determined by predicted rainfalls.

### 2.5.2   Basin sub-sampling for the climatological ensemble runs

Implementation of climatological ensemble predictions is numerically demanding: 3 lead times, 5 assimilation configurations, 60 seeds, 18 members, and numerous trials for model hyperparameter searching and training. To keep reasonable computation times, the climatological ensemble runs were conducted on a subset of 56 basins of the initial set of 531 basins. This subset of
basins was selected uniformly, according to their NSE rank from Kratzert et al. (2019) covering the same range of basins as the initial sample of 531 basins: see Fig. 6.

### 2.5.3   Softwares and hyperparameter settings

For the orchestrator (MLP) configurations, the hyperparameters listed in Table2 were optimized by exhaustive grid search and cross-validation. The hyperparameter subset was derived from a larger space using 20 randomly selected basins, retaining the
most frequent configuration. The hidden sizes ranged from a single layer of 30 neurons to four layers with multiples of 30 neurons. Five levels of learning rates ($10^{-1}$ to $10^{-5}$) were tested primarily, and two have been maintained according to their occurrences as best values.





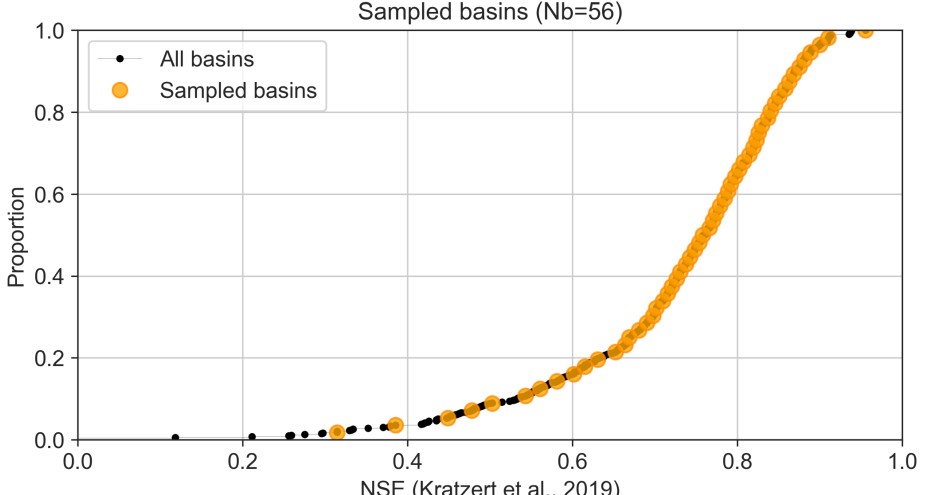

**Figure 6.** Distribution of NSE scores for trained LSTM models for the data sample used by Kratzert et al. (2019) and the selected subset of 56 basins

**Table 2.** Model hyper-parameter setup

| Parameters | Parameter space |
| --- | --- |
| Hidden layers [size,] | [120, 90] [120, 90, 60] |
| Activation | [relu, tanh] |
| Learning rate | [0.01, 0.001] |
| Solver | ADAM |
| Early-stopping | True |
| No_iter_no_change | 15 |
| Validation_fraction | [0.2] |
| Random seed | 60 |
| Sequence depth | 30 on forcings, 10 on assimilated data |

The experiments developed in this study are essentially based on open-source software and on the Python 3.9 programming language (van Rossum, 1995). Our modeling framework is based on the Scikit-Learn library (Pedregosa et al., 2012). Data analysis, processing, and visualization are performed mainly using Pandas (McKinney, 2010) Numpy (Walt et al., 2011), seaborn (Michael L., 2021), matplotlib (Hunter, 2007) and xskillscore (Bell et al., 2021). The model development was carried out using Jupyter Notebook (Kluyver et al., 2016), Anaconda (Anon, 2020), and PyCharm (JetBrains, 2024).





# 3   Results

The performance of the 3 DA approaches is compared with those of the benchmark models across the two forecast scenarios
tested. This choice emphasizes the contrast in model performance between an idealized context (perfect forecast scenario)
and a highly uncertain one (climatology-based forecast). Empirical Cumulative Distribution (ECDF), boxplots, and error bars
are used to show the variability of the scores across the basins. Different colors and line styles are used to distinguish the
model types and the approaches. The simple MLP appears in black; the benchmark SAC-SMA and LSTM are represented,
respectively, by blue and red dotted lines; color variations (blue to violet and red to orange) are used for the forecasting models
based on the benchmarks SAC-SMA and LSTM and including data assimilation. All presented results concern the test set.

## 3.1   Performances of the DA approaches based on perfect meteorological forecasts

### 3.1.1   Efficiency

Figure 7 shows the ECDF distribution of the persistency scores (PERS) for all approaches and lead times ($hp$) tested in the 531
basins.

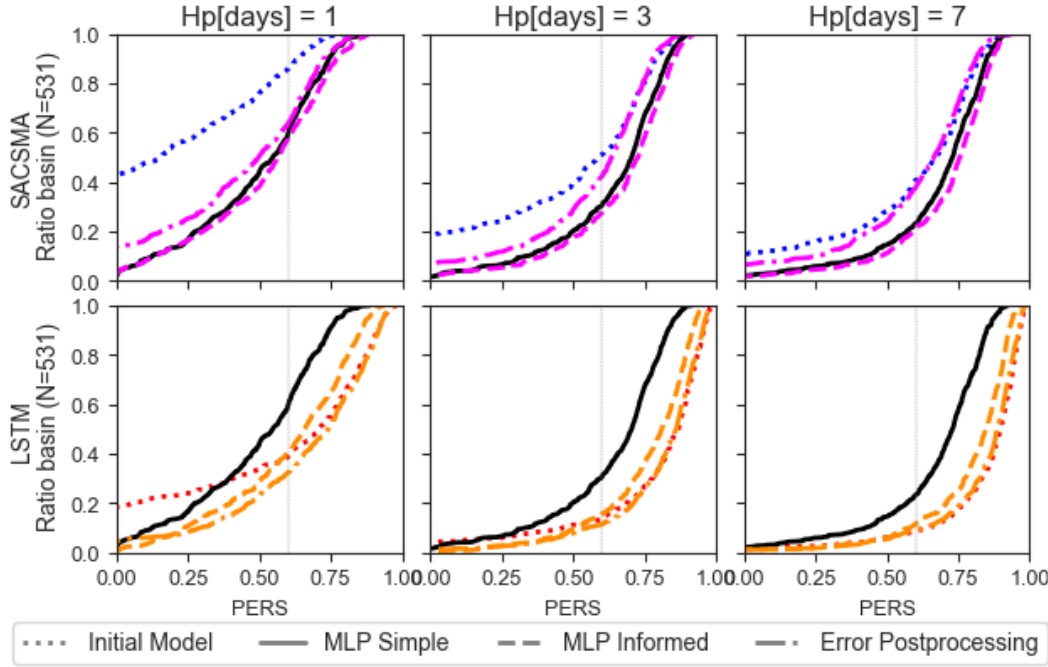

**Figure 7.** ECDF of the persistence scores. Rows are related to the benchmark cases (SACSMA and LSTM); columns are related to lead times
(1-,3-, 7-days). The baseline (MLP simple) is replicated in the rows. In the legend: *Initial Model* stands for the benchmarks, *MLP Informed*
means informed by the benchmarks, *Error Postprocessing* concerns the errors of the benchmarks models. Rightward-shifted curves indicate
better performances.




Without surprise, the PERS scores are lower in the short lead times. This is a usual result in persistence analysis since models generally struggle to beat the persistent model at very short lead times: the smaller the discharge variation, the harder it is to predict. Secondly, in line with previous studies, the LSTM model outperforms SACSMA. This remains valid even when the models are combined with data assimilation procedures.

The simple MLP-based data assimilation method appears more efficient than the benchmark SACSMA in all tested lead
times; it only outperforms the benchmark LSTM model in the lead time of 1-day, especially when the LSTM leads to negative PERS, which is observed for 20% of the basins for the LSTM model and less than 5% of the basins for the simple MLP. If we recall that, unlike the tested MLP, the LSTM model does not account for the past observed discharge, this first result highlights, as in numerous previous publications, the outstanding performance of the LSTM models in RR simulation.

The two proposed data assimilation procedures, based on the benchmark models (i.e., MLP-informed and error postpro-
cessing), prove to be effective, as they consistently improve the performance of the forecasting models they build upon. The MLP-informed model outperforms the simple MLP, while the error postprocessing approach generally enhances the persistence of the benchmark model, or at the very least, maintains its performance when it is already high. Figure 8 showing the gain in PERS of the various data assimilation procedures tested, compared to the benchmark models, confirms this analysis. The changes observed in the distributions in figure 7, correspond to systematic improvements in basins where the benchmark
model initially had lower scores. The error post-processing approach leads to positive gains or average null gains in cases where the benchmark model had initially high scores.

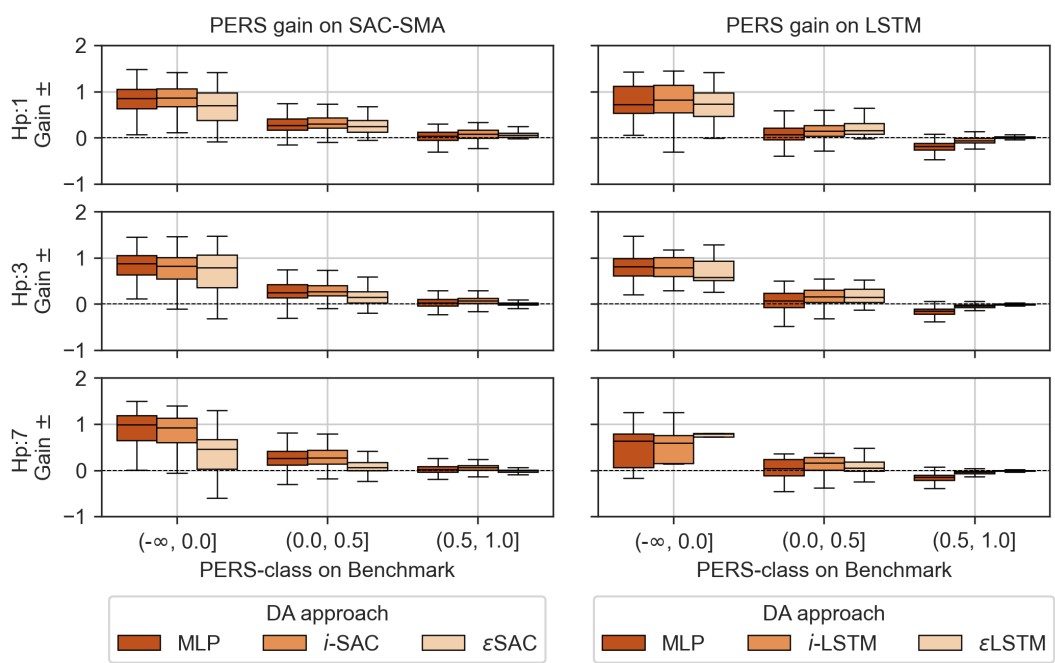

**Figure 8.** Gain on Persistence. Where, Gain= DA - Benchmark. For lighter nomenclature, the following names have been respectively used: MLP simple (MLP), MLP Informed by BM ($i$-SAC or $i$-LSTM), benchmarks error post-processed ($\epsilon$SAC or $\epsilon$LSTM)





Finally, the global ranking of all the tested approaches depends on the initial skills of the benchmark and simple MLP models. When the benchmark model is SAC-SMA, it is outperformed by both the simple MLP and the MLP-informed approach, but the latter appears to be the most efficient. However, the error post-processing method, based on the LSTM forecasts, is the most efficient of the approaches.

These first results demonstrate the effectiveness of the proposed DA strategies in improving the efficiency of the forecast in the perfect meteorological forecast scenario. Gains are particularly significant for the 1-day lead times. The added value of the proposed data assimilation methods drops rapidly with increasing lead times. This is to be related to the overall short response times of the basins included in the CAMELS-US data set, typically 1 to a few days according to the cross-correlation analysis.

Let us examine now whether these conclusions remain valid when considering uncertainties in meteorological forecasting.

## 3.2 Performances of the DA approaches under the climatology-based forecast scenario

As a reminder, in the climatology-based scenario, historical weather records of the past years are used as surrogate forecasts (i.e. highly uncertain forecasts). According to the sampling method used, every meteorological sequence, starting on the same date, recorded during the past 18 years of the training set, becomes one of the members of the $N = 18$ meteorological forecast ensemble for the 2-year test set. Combined with the $M$ random seeds of the trained model (8 for the LSTM benchmark, 10 for the SAC-SMA, and 60 for the other approaches), the ensemble forecasts count $N * M$ members. Three properties of these ensemble forecasts will be evaluated: (1) the efficiency based on the CRPS score, (2) their reliability based on rank diagrams complemented with spread/skill ratios, and (3) their resolution using Brier and AUC scores.

### 3.2.1 Forecast efficiency

Figure 9 shows the CRPS for the climatology-based scenario for all the approaches and lead times tested. The mean absolute error of the persistence model is included in this figure to maintain consistency with the reference method used in the previous section to evaluate the efficiency of the forecasting approaches tested.

Firstly, and encouragingly, most of the tested models remain more effective than the baseline persistence model, even when accounting for uncertainties in meteorological forecasts. Moreover, the performance gap between these models and the simple no-forecast persistence model widens as the forecast lead time increases. In other words, uncertainty does not negate the added value or efficiency of the forecasts.

Secondly, some trends are consistent with the results obtained with the PERS criterion and the "perfect" meteorological forecasts (Fig. 7). The proposed data assimilation approaches remain effective, as they improve the efficiency of the forecasting models they build upon, or at least do not significantly reduce it: the ECDF of the simple and informed MLPs are almost superimposed. Moreover, the benefits of data assimilation approaches are more significant in the shorter lead times and for the SAC-SMA benchmark model. However, the overall rankings of the forecasting methods tested have been completely overturned. The simple MLP model now appears to be the most efficient forecasting model in this ensemble forecasting exercise. Note that it is well-known that ensemble scores may be affected by the number of members of the evaluated ensembles (Leutbecher, 2019). Since the numbers of members for the benchmark models and the DA approaches differ, it has been verified




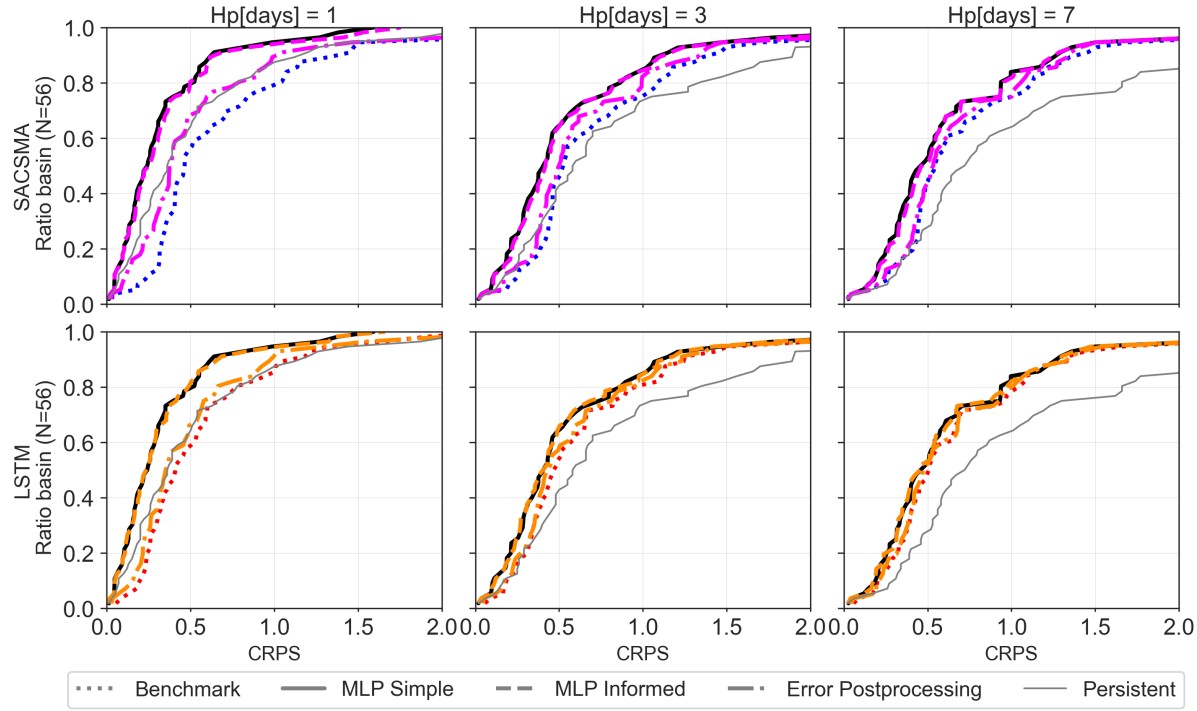

**Figure 9.** Empirical cumulative distribution functions (ECDF) of CRPS scores for the 56 test basins. Recall that CRPS=0 is the perfect model. Benchmark-related cases (SAC-SMA, LSTM) are presented in rows, lead times 1-, 3-, 7-days) in columns. Both the baseline (MLP simple) and the persistent model are replicated in the rows, the former is represented with bold dark line, and the later with a tiny gray curve. In the legend: *Benchmark* stands for the SAC-SMA or LSTM, *MLP Informed* means informed by the benchmarks, *Error Postprocessing* concerns the errors of the benchmarks models. Up- and Leftward-shifted curves indicate better performances.

that the computed CRPS values were not too much affected by this difference (see Appendix A). By reducing the size of the ensemble initialization to 8 seeds for the DA approaches in Figure 9, the distribution remains almost unchanged. This outcome ranking of the forecast models is robust, although it may seem surprising at first sight. The detailed analysis of the other properties of the forecast ensembles provides some explanation.

### 3.2.2 Forecast reliability

Figure 10 shows the rank diagrams for the climatology-based scenario: The ensemble members have been grouped into 10 classes for all models to facilitate comparison. The charts are organized vertically, and results for the two benchmark models are shown first, followed by the five DA approaches tested.





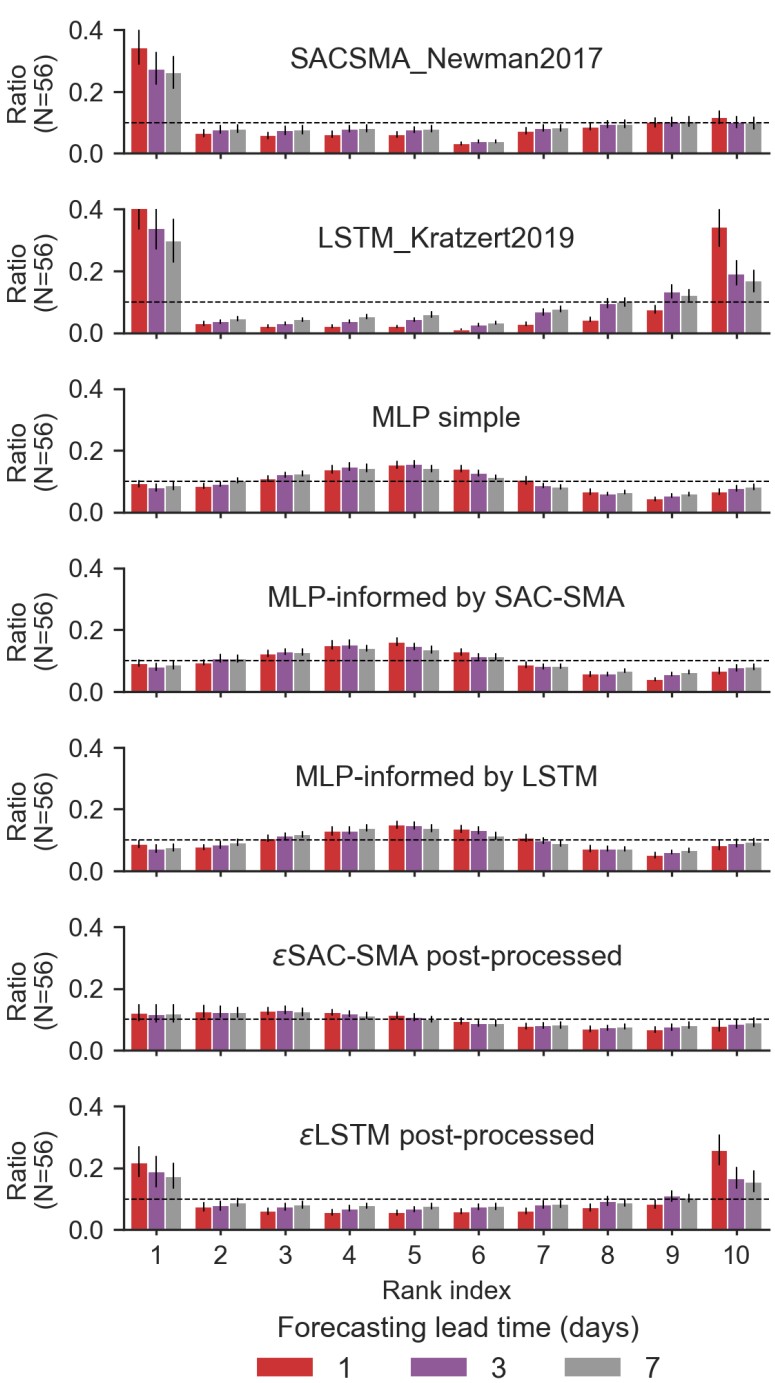

**Figure 10.** Rank diagrams for the benchmark models and the DA strategies. X-axis (10 rank classes), Y-axis (proportion of observed values in each class), median ratio and error-bars indicating the maximum and minimum ratios for the 56 test basins. Colors indicate the lead times.





Reliable forecasts are expected to produce flat, uniform rank diagrams, indicating that observations are evenly distributed across the range covered by forecast ensemble members. The most striking deviation from this ideal is observed in the U-shaped rank diagram of the LSTM model. When driven by meteorological ensembles, the LSTM simulations significantly underestimate forecast uncertainty, resulting in under-dispersed ensemble outputs. A disproportionately large number of observed values hence fall outside the ensemble range (ranks 1 and 10), indicating poor reliability. Figure 11, confirms that the underdispersion of LSTM ensemble forecasts is systematic. The overall spread of the LSTM ensembles is notably low compared to the root mean square error (RMSE) calculated from the ensemble mean. This discrepancy is particularly pronounced at the 1-day lead time.

This result suggests that the LSTM model is insufficiently responsive to recent meteorological inputs (i.e. corresponding to the meteorological ensembles) and is overly influenced by longer-term historical data. Although this characteristic likely contributes to its strong performance in deterministic RR simulations, it becomes a limitation in the context of ensemble forecasting. The error post-processing approach corrects this behavior, but only partly, for this dispersion bias of the LSTM ensemble forecast.

The SAC-SMA model does not exhibit the same limitations as the LSTM model, but a clear tendency to overestimate the discharges, and this tendency concerns all the 56 basins as revealed by the error bars. This bias is eliminated by the error post-processing approach.

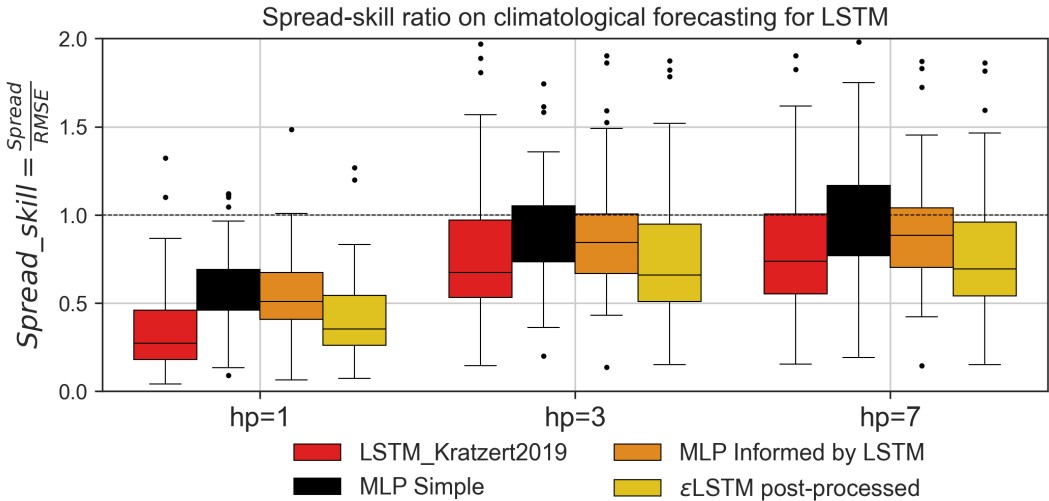

**Figure 11.** Boxplots of the spread-skill ratios on the climatology-based scenario for the 56 test-basins

To be efficient, an ensemble forecast should be reliable; and to be reliable, the observations should be well calibrated by the ensemble forecast. The LSTM forecasts seem to be hampered by the lack of reliability of the LSTM ensembles, and this explains the overturn of results between the deterministic and ensemble forecast scenarios. The MLP and MLP informed models exhibit a slight over-dispersion that appears to be less impactful to the model's efficiency. Does the efficiency reflect the




event detection capacity of the tested models ? Discharge forecasting models are often implemented operationally to predict
that a discharge or water level thresholds will be exceeded. To evaluate this capability, we will analyze the forecast resolution
of the proposed approaches in the next and last section.

### 3.2.3 Forecast resolution

The Brier score (Fig. 12) as well as the AUC (Fig. 13) confirm the classification of the forecasting models provided by the
CRPS. The LSTM performs globally much better than the SAC-SMA model. This is particularly clear in figure 12. DA methods
improve the resolution of the predictions compared to the initial benchmark models, while the simple MLP model provides
the highest skill. The proposed DA assimilation methods appear effective in the sense that they improve the skills (benchmark
versus error post-processing) or at least do not degrade too much the performance of the model they are based upon (MLP
simple versus MLP informed). It is important to recall that the optimal value is 0 for the Brier score and 1 for the AUC. The
classification of the models remains valid for the entire range of discharge threshold values and all test basins, as indicated by
the error bars in both figures. Both figures also indicate that all tested models have better resolution skills than the reference
random detection model, which theoretical value is equal to 0.25 and 0.5 for the Brier score and AUC respectively, and is
indicated by a red dotted line in the two figures.

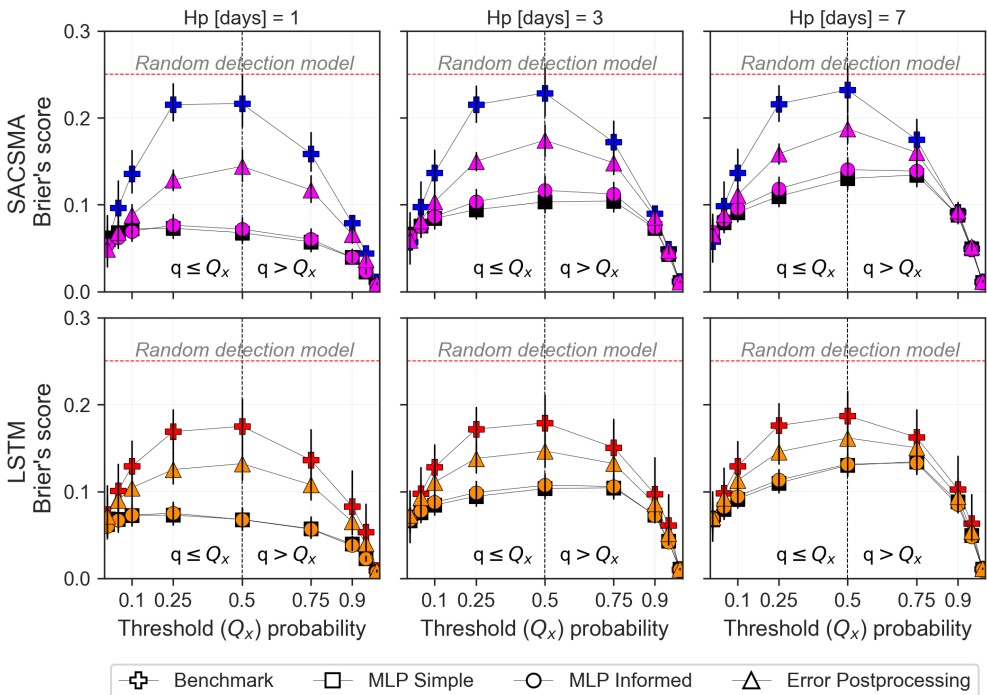

**Figure 12.** Brier Score for event detection thresholds based on discharge quantiles with non-exceedance probability [0.01, 0.05, 0.25, 0.5, 0.75, 0.9, 0.95, 0.99]. Median scores and error bars indicating the maximum and minimum scores across the subset of 56 test basins.



Although being globally consistent, both scores considered provide different insights into the relative resolution skills of the models tested, especially depending on the threshold level.

The Brier score evaluates how reliable the probability of threshold exceedance computed by the ensemble forecasting model
is. Since discharge values are varying seasonally and are highly auto-correlated, the probability of exceedance is high if the threshold value is already exceeded for high or low thresholds: i.e. the probability that the discharge will remain low (resp. high) is high during the low (resp. high) flow period. Therefore, the Brier scores have a tendency to converge towards 0 when the threshold values considered correspond to high or low quantiles of the considered series for all models. The Brier score is not a very discriminating criterion for such high or low thresholds. It appears to be better suited for threshold values corresponding
to the median range of observed values (Fig. 12).

Figure 13 shows the area under the ROC curves (AUC) for event detection, based on various discharge thresholds covering both, flood and drought forecasting. An example of the corresponding ROC curve, for the threshold probability of $q > Q.95$, is reported in Appendix B1. Low and high discharge detection performances are displayed, respectively, on the left and right sides of each subplot, and marked, respectively, with $\leq$ and $>$. The benchmark cases are organized in rows, and the lead times
370 in columns. Colors and markers are used to differentiate the 5 DA approaches. The median AUC values are shown as well as the spread of values over the 56 basins.

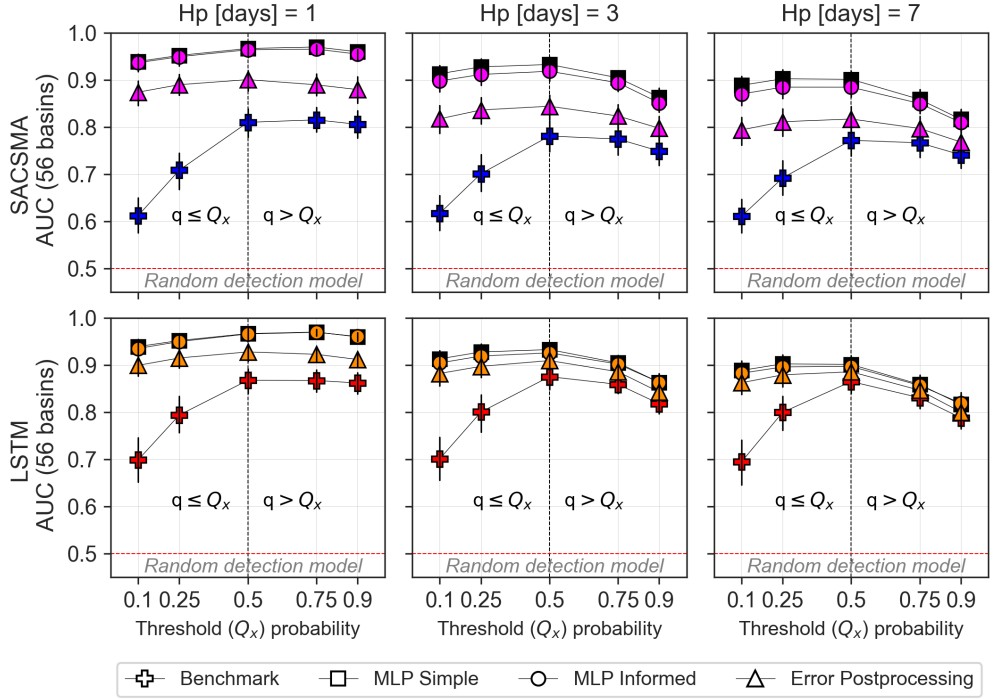

**Figure 13.** AUC values for events based on flow quantile [0.1, 0.25, 0.5, 0.75, 0.9], with drought/flood detection shifting at quantile 0.5. These are the median AUC values across 56 basins. Colors highlight benchmarks (SACSMA: blue to violet, LSTM: red to orange), with the baseline MLP-Simple replicated in black.





The AUC, calculated based on ROC curves (see Fig. B1 in the appendix), measures how accurate the balance between the probability of detections (POD) and false alarm ratios (FAR) is for ensemble forecasts. It offers a clearer contrast of the models' resolution skills over the whole range of discharge thresholds (Fig. 13). The differences between models are larger for low thresholds (i.e. low flows). It is necessary to recall the impact of the mean squared error (mse)-based loss functions on model adjustment, as they fundamentally favor the high values in the target variables. This may explain why the AUC values are significantly low in the lower flow ranges for the benchmark models. However, this limitation looks improved by DA procedure, as these scores are increased on these low discharge thresholds. The AUC has a tendency to decrease with increasing discharge thresholds for longer forecasting lead times (3 and 7 days). Likewise, the added value of the DA methods compared to the LSTM benchmark for these higher thresholds and longer lead times aligns with the results of the deterministic forecasts: the added value of the DA strategies is limited. For lower discharge thresholds, the improvements from DA procedures remain substantial, even at extended lead times. Two factors likely explain this result. Firstly, the benchmark models have less skill in predicting low flows, which were not the primary focus during training, leaving more room for improvement. Secondly, low flows tend to exhibit a much longer autocorrelation range than short-lived flood events associated with high flows, which explains the relevance of DA methods for longer lead times for this range of discharges.

## 4 Conclusions

This work aimed to evaluate the added value of data assimilation procedures to rainfall-runoff forecasts, especially in the context of AI-based LSTM forecasts. The two proposed evaluation frameworks yield contrasted results. Figure 14 shows the results of some of the forecast methods evaluated, for one flood event, in a test basin, and the two forecast configurations. It is provided here for illustrative purposes, providing additional insights into the results beyond the global evaluation criteria. Of course, general conclusions should not be drawn from this isolated example.



**Figure 14.** Illustration of various forecasts of the same event: perfect meteorological forecast (left) and climatological ensemble forecasts (right). the orange dots are the observed discharges. The halos represent the confidence intervals (CI: 100%, 99%, 95%, 90%, 80 %) of the ensemble forecast.

The main conclusions of this study are the following. The various data assimilation procedures tested appear effective, as they generally improve - or at least do not significantly degrade - the forecasting performance of the models on which they are based. Under the hypothesis of perfect meteorological forecasts, the DA approaches consistently improve the SAC-SMA forecast, while the gains are observed mainly for short lead times and basins where the benchmark LSTM model initially





underperformed. These limited improvements confirm the strong performance of the LSTM model in rainfall-runoff simulation and forecasting, as already demonstrated by numerous publications.

However, this conclusion is entirely overturned in the context of the climatological ensemble evaluation framework. The simple MLP model, which incorporates past observed discharges, outperforms the LSTM model even when the latter is coupled with DA assimilation procedures, at least for short lead times. In this setting, the LSTM model is penalized by the unreliability, specifically the under-dispersion, of its forecast ensembles. Its predictions appear insufficiently responsive to meteorological forcing over the forecast horizon. Of course, while climatological ensembles are likely more spread than the actual meteorological ensemble that may be available in practice, the proposed ensemble evaluation framework proposed herein may appear conservative. However, this finding highlights the need to ensure the reliability of LSTM ensemble forecasts for their effective operational deployment.

For high discharge forecasts, the added value of data assimilation (DA) procedures appears to be limited to lead times on the order of the basin's time of concentration. In the case of low-flow forecasting, Figure 13 suggests that: (1) DA could remain beneficial over longer lead times, and (2) forecasting performance could be significantly enhanced by using models specifically trained for this discharge range. The use of mean square error as the default training criterion in most AI packages tends to prioritize accurate fitting in the higher range of discharge values, often at the expense of low-flow performance.

Finally, the DA methods tested here were implemented with relatively simple MLP models. Although this choice aligns with the goal of developing frugal AI solutions, there is undoubtedly scope for improvement by exploring more advanced AI techniques in future works. Further research may also explore alternative ensemble forecasting strategies, such as the use of forecast archives or more sophisticated ensemble selection methods.

*Code and data availability.* TEXT

All data used in this study are sampled from the CAMELS-US dataset, available at https://gdex.ucar.edu/dataset/camels.html. The processed version of these data, prepared for this study, is archived at https://doi.org/10.5281/zenodo.16944643 , with detailed instructions provided both in their roots and in the model repository MLP_REPO. This repository typically contains the codes for the orchestrator presented above. The adapted benchmark models are available at LSTM and SACSMA. The original benchmark models are described in their respective publications and should be consulted prior to using these adapted versions. The post-processing code may be made available upon justified demand.





## Appendix A: Appendix

### A1 Impact of the number of members of the ensembles

Figures A1, A2, and A3 show that the influence of the number of members considered in the forecasts of the DA ensemble
is very limited. Figure A1 shows in particular that the relative increase or decrease in the CRPS value is negligible when the
number of members of the ensemble is reduced.

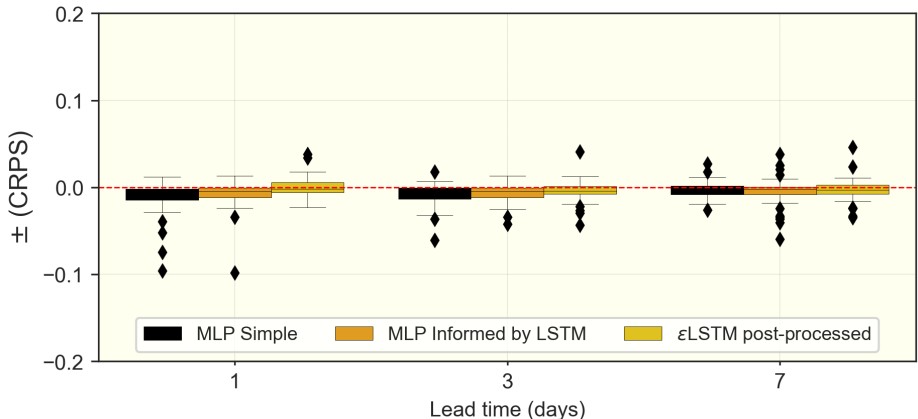

**Figure A1.** relative increase or decrease in CRPS scores $(CRPS_{1080} - CRPS_{144})/CRPS_{1080}$ when considering $N$ members =144 (8x18)
rather than $N$ members =1080 (60x18) in the DA ensemble forecasts.

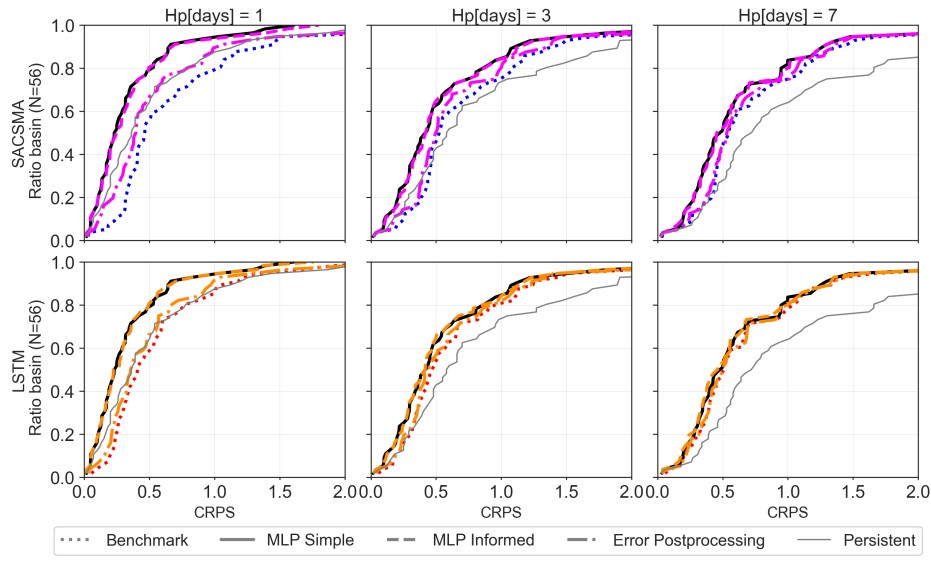

**Figure A2.** Same as figure 9: CRPS with 8 seeds (N members =144) rather than 60 seeds (N members =1080) for the DA ensemble forecasts.



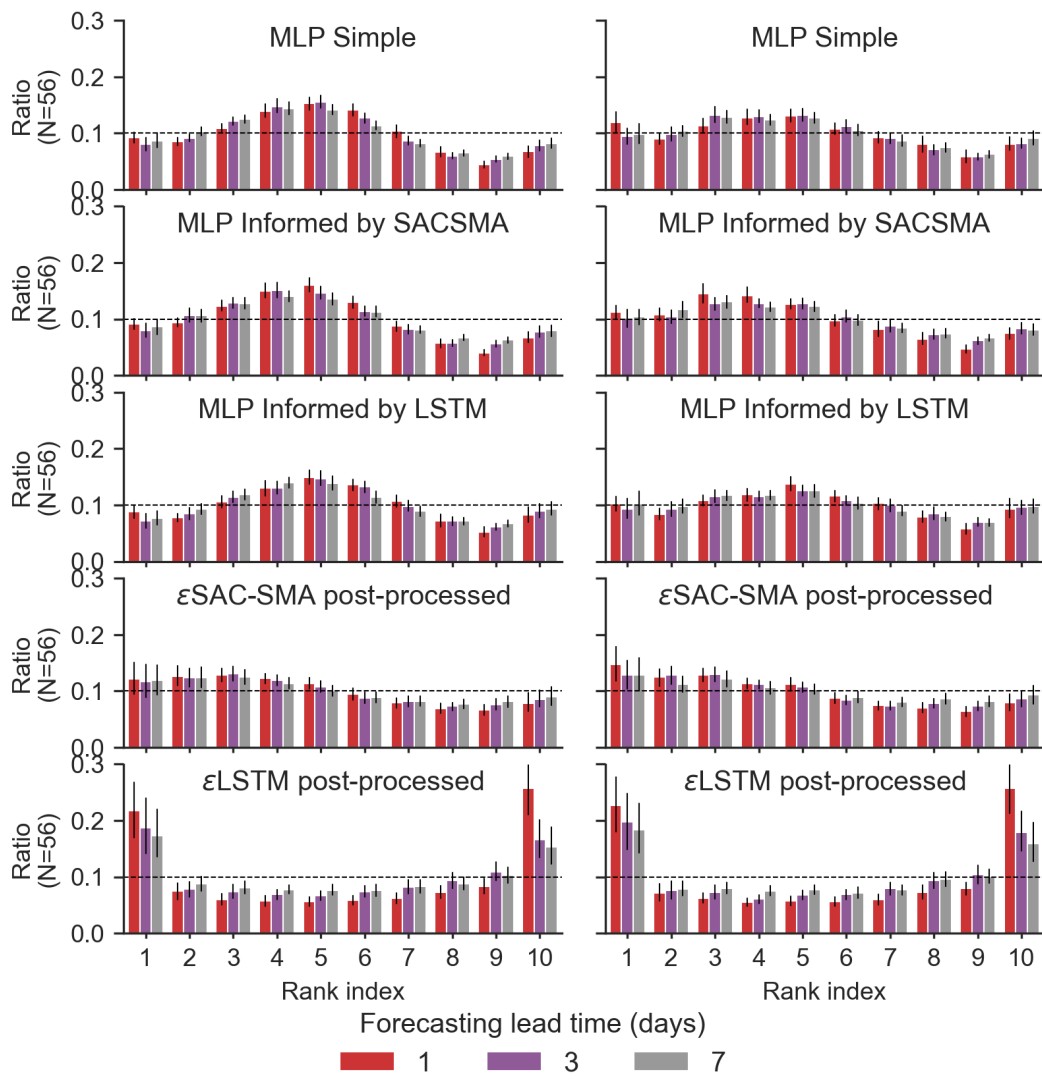

**Figure A3.** Comparison of the rank diagrams obtained with various ensemble members: N=1080 (left) and N=144 (right).

## B1 Illustration of ROC curves

Figure B1 provides an illustration of the ROC curves based on which the AUC values have been calculated, as well as the variability of the ROC curve shapes across the 56 test basins. One ROC curve and one AUC value are computed for each basin
and each forecasting method tested.




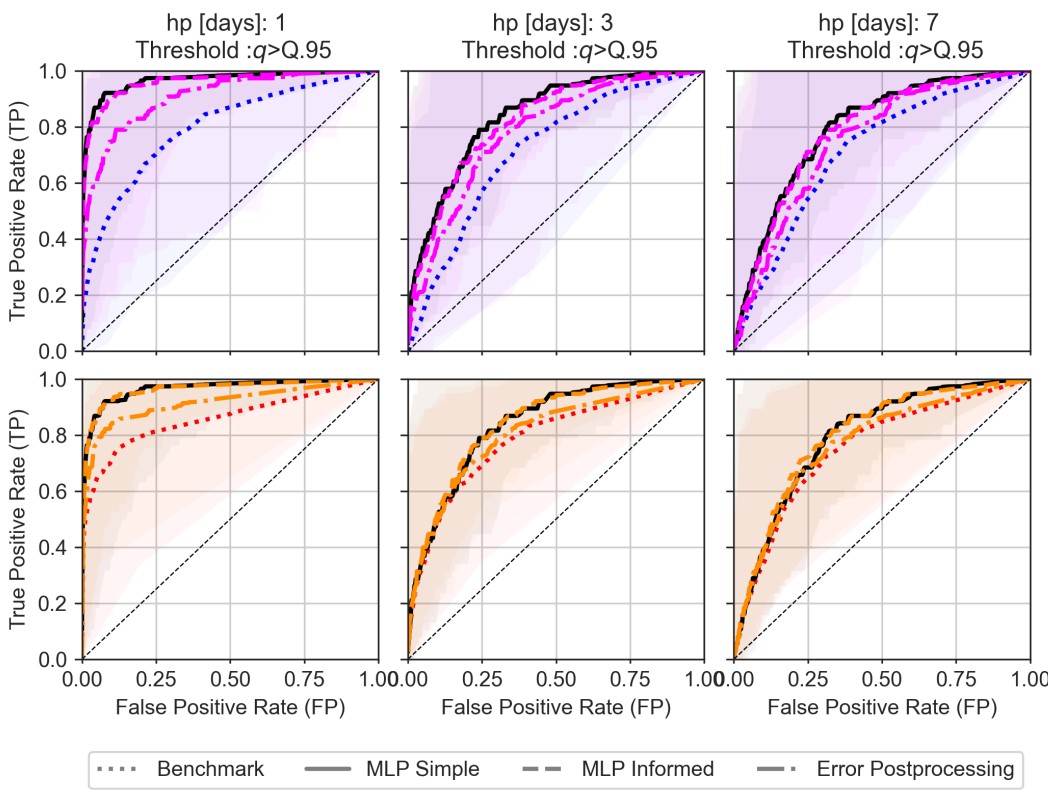

**Figure B1.** ROC curve for flood detection ($q \geq Q.95$) for 1-, 3- and 7-days lead times. Results are style-coded: **MLP Simple** (dark solid, DA-1), **MLP informed by benchmark** (dashed, DA-2), **Benchmark ePP** (dot-dashed, DA-3), **initial Benchmark** (dotted). Benchmark cases are color-coded: **SACSMA** (blue to pink, first row) , **LSTM** (red to orange, second row). Halos show the variability across the 56 basins around the median values.



*Author contributions.* All the indicated authors contributed to the realization and the discussions of this study. BSF and EG carried out the experiments and the analysis of the scientific relevance of the results. BSF developed the model code, performed the simulations and post-processed the results. FS participated in the deployment of the SAC-SMA model, including the post-processing of the results. NA and DT contributed in the discussion for the operationalization of the models as the aQuasys partners.

*Competing interests.* The authors declare that they have no conflict of interests

*Disclaimer.* The paper is written in LaTeX using **Overleaf**. **Writefull** and **ChatGPT** have been used for rephrasing and minor corrections. The experiments are essentially based on the CAMELS dataset and open-source software and languages such as Python 3.9, scikit-learn, numpy, pandas, ....

*Acknowledgements.* The authors would like to thank **Gustave Eiffel University** and **aQuasys Company** for bringing together the AI_Eau
project, funded by the ***Agence nationale de la Recherche (ANR)*** under the ***France 2030*** program. We are grateful to the NeuralHydrology team for making their regional LSTM code publicly available, as well as the authors of the SAC-SMA model and the CAMELS dataset. We also acknowledge the ***Centre de Calcul Intensif des Pays de la Loire (CCIPL)*** for providing the computing resources. Finally, we thank Zoë Jack for her valuable contribution with preliminary proofreading and grammar checking.





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
