# Peer review of "Testing data assimilation strategies to enhance short-range AI-based discharge forecasts"

_EGUsphere, 2025_

## Author Comment (AC1)

Replies to https://doi.org/10.5194/egusphere-2025-4244-RC1 for the preprint https://doi.org/10.5194/egusphere-2025-4244, by Saint Fleur et al. (2025)

We would like to thank the reviewer for these very constructive and stimulating comments which will help improving the initial version of the manuscript. We provide replies and explanations to the reviewer's questions in the following.

**Comment 2.1**

The authors indicate that they are benchmarking against Kratzert2019 and Newman2017, but they have different training / testing periods as the benchmarks. In section 2.1 you indicate that you are training from 1989-2006 and testing in 2006-2008. However, Kratzert used 1999-2008 for training and 1989-1999 for testing. This means you are training much more than them, and testing in only a 2-year period. Why this difference?

Moreover, you are also indicating that you re-simulated the 1989-2008 period with the pretrained models from Kratzert and Newman, but again, you are not respecting the training/testing split that the authors did in their original study.

Using benchmarks is an extremely valuable technique, because it automatically places your method in existing literature, however the conditions of the original studies need to be respected. One should, if possible, adapt the new experiment to the existing benchmark, otherwise it does not make any sense to do a benchmark.

We thank the reviewer for this important comment. Using training and testing periods that differ from those in previous studies complicates direct comparisons. As pointed by the reviewer, the selected periods also imply that all approaches in our manuscript were not evaluated under identical conditions. Our work is built on Kratzert et al. (2019) and Newman et al. (2017), those reference models were trained and tested using matched periods. In our current setup, the 2006-2008 period used to evaluate the data assimilation (DA) strategies lies within the training period of the reference LSTM and SAC-SMA models. This choice inflates the apparent performance of the baseline models and likely disadvantages the DA approaches.

In the revised manuscript, we will align the training/testing split across all the approaches, using 1999-2008 for training and 1989-1999 for testing, ensuring consistent comparison.

Also, why are you using only two years of testing?

Our initial choice was to maximize the size of the training set, following common practice in AI-based studies, at the expense of a shorter testing period. Although a smaller testing set may increase the influence of sampling variability on the results, this is partially compensated

by the large number of basins in the test data set. Nevertheless, in light of the previous comment, we will revise the training/testing split in the updated version of the manuscript.

Lastly, you should benchmark your study against other studies that used data assimilation methods. In CAMELS US there is the study of Nearing (2022), Feng (2020) or more recently Yang (2025).

We thank the reviewer for bringing these studies to our attention. References to Nearing (2022), Feng (2020) and Yang (2025) will be considered in the revised manuscript. Pending further verification, some of the approaches proposed in Nearing (2022), as well as other relevant studies, will be included in our comparative analysis.

**Comment 2.3**

In section 2.3 you indicate that a specific model should be calibrated for each lead time, and that the alternative is inefficient. But this is not true. The LSTM has a linear layer at the end that transforms the hidden states to discharge.

Your model is a simple extension of this, but instead of a linear layer that go from hidden states to discharge, you have a feed forward neural network (so a couple of linear layers), in which, besides the simulated discharge you concatenate some past meteorological variables and discharge. I do not have anything against the simplicity of the model, because if it is simple and it works then great, but there are multiple things that need to be considered.

You can run the LSTM as a seq-seq model, roll it over the forecast horizon (so if you are predicting 7 days forecast just do seq-7) and in each step concatenate the hidden states with whatever you want and pass it through the feed forward neural network. This way you have a consistent and generalizable model that does not require a different embedding for each case.

The alternative we had in mind is the recursive forecasting approach. The reviewer is correct, the seq-to-seq architecture is another alternative which also introduces additional challenges, particularly the risk of obtaining intermediate performance across lead times. Training may become more complex and prone to sub-optimal solutions for individual lead times. Nonetheless, as shown in Nearing (2022), seq-to-seq models remain an interesting option in our perspective.

Also why are you evaluating only on days 1, 3 and 7? Why not all seven days?

This choice was for time saving reasons, the results for the intermediate lead times are expected to evolve monotonically. However, to avoid ambiguity, results for the complete range of the 7 days lead time will be included in the revised version.

Furthermore, you are using climatological ensembles to create a possible forecast, but I do not believe this is the best strategy to do that. Climatology is used, normally, in medium to seasonal range (so couple of weeks to some months) where the forecast models of the meteorological variables are no longer reliable, and for variables that present a cyclic pattern (temperature, radiation…)  but you are using it in precipitation 1 or 3 days ahead, which I do not believe is a good practice. How is the precipitation of the 1st of November for the last 18 years related to the 1st of November of this year?  I do not think there is a strong relationship between these values that can be used for short-term forecasting, especially for precipitation, which is the most important variable to drive the forecast. If you want to use it for temperature or radiation, that can be an (non-ideal but defendable) option, but I would highly recommend to not use it for precipitation.

The use of climatological ensemble as a no-skill benchmark for meteorological forecasts is a common practice to evaluate hydrometeorological forecast performances.  Figure 4 illustrates that the climatological ensembles distribution does not reveal any evident bias for both the rainfall and the PET despite the simplicity of their construction.

In this section, you also indicate the absence of operational weather forecast archives, but this is also not fully correct. Shalev (2024) release a historical weather forecast for CARAVAN, which includes the CAMELS US dataset. It is true that not all the products are available in the testing periods that you have but:  One option is to use CHIRPS-GEFS which is only precipitation but is available in your period of interest. Another, better option, is to benchmark a simple LSTM against Kratzert 2019, and once that is working well, you evaluate your model and the new LSTM in the periods where you have a historical forecast (2016-2024) from Shalev. This is more work but would actually give you a robust study to evaluate your model under real forecast conditions.

We thank the reviewer for mentioning this study and the associated datasets. Nevertheless, the use of the meteorological forecasts (and nowcast) products provided by Shalev and Kratzert (2024) in the present study presents major challenges, several of which were identified by the reviewer. First, CHIRPS-GEFS provides only precipitation forecasts and covers the training but not the testing period that should be used in the revised version of the manuscript. Its integration would require generating realistic forecast scenarios for the remaining forcing variables, applying an appropriate bias-correction procedure to precipitation and rerunning the entire training/testing workflow process for all tested models. In addition, no overlap period exists between the meteorological forcing series used in the present study (Maurer extended version) and the historical forecasts products available on the 2016-2024 period. **Figure A** summarizes the availability of the

variables between the products suggested by the reviewer and the datasets used in the present work.

[Figure]

*Figure A: Availability between the needed archives and those provided in Shalev and Kratzert (2024). Screenshot are shown for the nowcast products (top-left), forecast products (top-right) up to our covered lead times, and an example of the Maurer extended forcing variables (bottom) which stops in 2008.*

For these reasons, the use of these meteorological forecast series appears as a research work in itself, beyond the scope of this manuscript. The text of the manuscript will be modified to mention the existence of these historical forecast databases and explain why they were not used in this work.

**Section 2.2 and 2.5.2**

Here you indicate that you use an ensemble of 60 runs. Why 60? Most studies use between 5 and 10. Do you get a significant different with 60? If you want to use 60 that is your choice, but in section 2.5.2 you indicate that because of the 60 ensembles you have unreasonable computational cost, and that you are only going to use a subset of basins. If 5 to 10 ensembles give you the same as 60, then you can reduce the computational costs, and then do the study in the full region, which would produce more robust result.

Also, the 18 ensembles members can be accommodated in the batch dimension of the tensor and the different seeds can be run in parallel (even in a single GPU) so most of the computational overhead you are reporting can be overcome with some technical tweaking.

We thank the reviewer for this insightful remark. Our initial choice to use a large number of seeds aimed at thoroughly representing the modeling uncertainty. This choice ultimately proved computationally demanding, particularly for the climatological ensemble evaluation. Following the reviewer's recommendation, we will limit the revised experiments to 20 seeds. However, we will keep the same sample of 56 basins for the climatological evaluation, as it remains sufficiently representative of the full set of 531 basins.

Line 38: I do not agree that discharge simulation and discharge forecasting are fundamentally different tasks. You are trying to model the same system and the same rainfall-runoff response. In forecasting mode, you have the increased uncertainty of the meteorological input, however that is more of a limitation and not a fundamentally different task. Multiple operational models are calibrated with observed data in pseudo-forecast mode and later incorporated in forecasting pipelines, and they work well. Data-driven methods give the advantage that, if trained with real forecast, they can learn to compensate for systematic biases, but again, this is more of training strategies to compensate for data quality limitations and not because the task is fundamentally different.

We thank the reviewer for this remark. We will revise this paragraph (lines 38-46) in the final version.

Line 66. The tittle of Fig 1 should be more self-explanatory.

The legend of Figure 1 will be modified: "Cumulative density functions of the Nash-Sutcliffe and persistence criterions obtained for two types of rainfall-runoff models - the conceptual model SACSMA (Newman et al., 2017) and the LSTM proposed by Kratzert et al. (2019). The

two models have been trained and evaluated on the CAMELS-US dataset. The illustrated scores correspond to the independent evaluation period."

Line 127-128: What do you mean by: "The direct forecasts from the benchmark models were assumed to be unchanged for the tested lead time; therefore, no further running was necessary".

To remove any ambiguity, this sentence will be reformulated as follows: "The forecasts of the initial models are used as inputs of the proposed DA approaches ".

Figure 7. You should explain the colors also on the legend of the figure and not only on the text above. The figure plus the legend should be self-explanatory. Also, as a suggestion the message of this figure would be better explain by a boxplot per lead time graph. The boxplot would give the distribution along the basins and because you have one for each model and for each lead time it can be easily compared. Something similar to your Figure 8, but for the different lead times (you can also see Figure 3 from Nearing 2024).

The same color will be used in all graphics in Figure 7. Moreover, Figures 7 and 8 present complementary illustrations of the results. ECDF-plots such as in Figure 7 are common in previous works and highlight differences between models, corresponding to shifts of the ECDF, more clearly than boxplots. The boxplots on the other hand, show to which extend the gains are systematically observed for all watersheds or not.

Section 3.2.2: Can you explain in detail how did you constructed this figure? How did you construct the 10 classes? What does a lower or a higher rank indicates?

The tested (DA) approaches differ in ensemble sizes, with some configurations using 18 x 8 members (forecast members and seeds) and others using 18x60 members. For the ease of comparison, all ensembles were reduced into 10 equiprobable classes for the rank diagrams.

Technically, the ranks of the observed discharge relative to the ensemble forecast were calculated using the *rank_histogram()* function from the **xskillscore** library, which is based on Hamill (2001). The so-obtained rank values were then grouped into 10 bins using the *groupby_bins()* method applied to the corresponding *xarray.DataArray* object, with the **bins** argument set to 10. Each bin was assigned a categorical label (1-10), and the frequency of occurrences in each category was computed over the evaluation period for each basin.

The lowest class represents the frequency of the observed discharge falling lower than the ensemble's $10^{th}$ expected percentile, whereas the highest class corresponds to observations exceeding the $90^{th}$ percentile.

To assess the potential impact of differences in ensemble size across approaches, we constructed an illustrative example based on a synthetic ensemble drawn from a Gaussian distribution N(0,1). Using 1000 realizations and 30 independent resampling, we compared the 10-class rank histograms obtained with ensembles of 100 and 2000 members. As shown in **Figure B**, and as expected, both ensemble sizes produce uniform distributed rank diagram.

[Figure]

[Figure]

*Figure B: Example of a 10-class rank diagram constructed from a Gaussian distribution (N (0,1)) generating an ensemble of 100 members (left) and 2000 members (right), with 1000 examples each. In each case, the last member (100th or 2000th) is used as the verification value. Error bars indicate variability across 30 independent random realizations, while uniformity line is shown with the horizontal dotted black line.*

Line 330-335:  Can you explain in detail how are you evaluating the LSTM here to produce these results?  Also, you are indicating that "This result suggests that the LSTM model is insufficiently responsive to recent meteorological inputs".  Can it be that because the LSTM is driven only by meteorology, and because the climatological forecasts are not good (see my comment above about that) then the predictions are biased? The other models have the advantage of having discharge, which is a highly autoregressive variable, so they somehow compensate. However, if all you have is meteorological forecast and these are non-sense, how can the model perform well? I think this point is important and can biased the results you are presenting.

We acknowledge the reviewer's concern; but none of the presented results indicate any significant bias, either in the meteorological ensembles or in the produced discharges forecast.

As illustrated in Figure 4, and further supported by the rank diagrams of the other tested models in Figure 10, the proposed climatological ensembles are widely spread (reflecting no-skill meteorological forecast) but remain neither biased nor of non-sense. Similarly, the rank diagrams (Figure 10) do not indicate any systematic bias in the prediction for any of the tested methods, including the LSTM. A minor bias, a slight overestimation, may be observed in the rank diagram of the MLP Simple approach. The observed differences in the model's skills are

therefore attributable not to forecast bias but to the reliability of the ensembles (i.e., of their spread), which inevitably affects the forecast resolution.

In the context of ensemble forecasting, the LSTM appears penalized when fed with no-skill ensembles due to the relatively low dispersion of its predicted ensemble members. This reduced spread, compared to the other approaches, is clearly visible on Figure 11 as revealed the rank diagrams in Figure 10.

It is also important to note that when meteorological forecasts have little or no skill, the model's performances rely primarily on the hydrological inertia of the watersheds, a factor accounted for in all tested approaches, including the original LSTM.

Finally, the paragraph in lines 330-335, which appears awkwardly written, will be reformulated or eliminated in the revised version.

**References**

Nearing, G. S., Klotz, D., Frame, J. M., Gauch, M., Gilon, O., Kratzert, F., Sampson, A. K., Shalev, G., & Nevo, S. (2022). Technical note: Data assimilation and autoregression for using near-real-time streamflow observations in long short-term memory networks. Hydrology and Earth System Sciences, 26(21), 5493–5513. https://doi.org/10.5194/hess-26-5493-2022

Nearing, G., Cohen, D., Dube, V. et al. Global prediction of extreme floods in ungauged watersheds. Nature 627, 559–563 (2024). https://doi.org/10.1038/s41586-024-07145-1

Feng, D., Fang, K., & Shen, C. (2020). Enhancing streamflow forecast and extracting insights using long-short term memory networks with data integration at continental scales. Water Resources Research, 56,e2019WR026793. https://doi.org/10.1029/2019WR026793

Yang, Y., Pan, M., Feng, D., Xiao, M., Dixon, T., Hartman, R., Shen, C., Song, Y., Sengupta, A., Delle Monache, L., & Ralph, F. M. (2025). Improving streamflow simulation through machine learning-powered data integration and its potential for forecasting in the Western U.S. Hydrology and Earth System Sciences, 29(20), 5453–5476. https://doi.org/10.5194/hess-29-5453-2025

Shalev, G., & Kratzert, F. (2024). Caravan Multi Met: Extending Caravan with multiple weather nowcasts and forecasts. arXiv preprint arXiv:2411.09459. https://arxiv.org/abs/2411.09459

Kratzert, F. (2019). CAMELS Extended Maurer Forcing Data, HydroShare, https://doi.org/10.4211/hs.17c896843cf940339c3c3496d0c1c077

Hamill, T. M., 2001: Interpretation of Rank Histograms for Verifying Ensemble Forecasts. Mon. Wea. Rev., 129, 550–560, https://doi.org/10.1175/1520-0493(2001)129<0550:IORHFV>2.0.CO;2.

Talagrand, O., R. Vautard, and B. Strauss, 1997: Evaluation of probabilistic prediction systems. Proceedings, ECMWF Workshop on Predictability, ECMWF, 1–25. [Available from ECMWF, Shinfield Park, Reading, Berkshire RG2 9AX, United Kingdom.].

---

## Author Comment (AC2)

Response to the review #2 of the manuscript: **AC** (in classic black) and **RC (in red)**)

"Testing data assimilation strategies to enhance short-range AI-based discharge forecasts" by Saint-Fleur at al.

Scope:

The manuscript is in the scope of the journal.

Summary:

The authors exploring two MLP-based data assimilation strategies to improve the discharge forecasts of two different hydrological models, an AI based LSTM and the mechanistic SAC-SMA model. The resulting forecasting setups are applied for three different lead times within two forecast scenarios, a deterministic optimal forecast and an ensemble forecast based on a climatological ensemble approach.

In my opinion the manuscript needs improvements in content and structure.

On behalf of all co-authors, we thank the reviewer for the time and effort devoted to reviewing this preprint including the insightful comments and questions provided.

General comments:

The manuscript focuses on DA strategies. But why do the authors choose exclusively MLP-based strategies? This choice is not sufficiently justified in the introduction. How would establish DA strategies such as Ensemble Kalman filters perform in comparison? Are there already studies that address this question? And what are the advantages of using MLP-based strategies over established DA strategies?

We thank the reviewer for these insightful questions. A comparison between the MLP-based strategies and the Ensemble Kalman Filter (EnKF) (Clark et al., 2008) would indeed be highly informative. However, implementing such a comparison in a consistent manner would require substantial additional methodological developments that go beyond the scope of the present study. The EnKF approach relies on perturbations of inputs and/or state variables, and potentially of model outputs, whereas the used DA strategies considered here are restricted to observed and simulated discharge only. Nevertheless, we consider this comprehensive comparison as promising direction for future work.

In this regard, it would also be necessary to examine the required amount of training data. Is 18 years really necessary for this? And are there any catchment-specific differences?

In light if this comment, including that of RC1, the training period is being reconsidered. Since the MLP assimilates the output of the benchmark models, only their evaluation period will be used for training. The analysis will mainly focus on the added value of the MLP as an orchestrator (or assimilator). A catchment-wise analysis is also a highly relevant perspective; however, this is seen for now as a promising investigation for future work.

The chosen training method also seems rather random. Why are 60, 8 and 10 seeds used? Would smaller numbers also be suitable? Why are the reference models treated differently?

The initial motivation for using such a large number of seeds was to obtain a sufficiently large ensemble for forecast analysis. However, preliminary results indicate that approximately 20 seeds are sufficient, and it will be reduced in the revised version of the manuscript.

In terms of structure: the authors use result figures in the introduction and conclusion, which in my opinion is not good style. On the one hand, they refer to content introduced later on, and on the other hand, they add new content when a summary is required. Figure 1 is unnecessary, as its content is repeated in Figure 7. Figure 14 shows examples of the analysed data and should be presented at the beginning of the results section to give an impression of the data. A similar figure should be presented for the SAC-SMA model at least in the appendix.

We thank the reviewer for this comment and will revise the structure of the manuscript accordingly.

Minor comments

Chapter 1 intro

- L38: I wouldn't argue that forecasting is "a fundamentally different task" since hydrologic discharge simulation models are used for forecasting but in a wider framework.

This sentence will be removed in the revised version, as the process does not differ between forecasting and simulation.

By "model structures", we refer to imperfections related to the design and configuration of the model (e.g. architectural choices and parameterization) which may lead to structural model errors.

- L55-65: manuscript structure, see comment above

Chapter 2 Materials and methods

2.1 dataset

- style issue: repeating citations are not necessary

Thank you for this remark, we will remove the repeated citations accordingly.

2.2 data assimilation

- L99-104: to be placed and elaborated in the introduction

We thank the reviewer for this suggestion, and we will restructure the manuscript to improve clarity and coherence in the revised version.

- L129-L134: difficult to understand, "In both forecasting approaches --> this is not clear. You describe three DA approaches. But is MLP alone a DA approach or rather an alternative ML approach compared to LSTM?

In this paper, the MLP is presented as an orchestrator, it is rather a DA approach than a model to be directly compared with the LSTM. The term "both forecasting approaches" refers specifically to the "deterministic" and the "ensemble" forecasting strategies. We will clarify this point in the revised version of the manuscript.

2.3. forecasting setup

- L154: climatological ensemble --> unusual for daily data, provide discussion and reference

The "climatological ensembles" approach employed in this study was considered as an alternative to the optimal "deterministic forecast" investigated. We have deliberately tested a "poor man's ensemble" by sampling all the meteorological variables on a dateto-date basis from past observations. While this approach is unusual and may appear counterintuitive for daily rainfall data, it nonetheless represents a valuable alternative. Furthermore, it is conceptually similar to the Ensemble Streamflow Prediction (ESP) approach proposed by Day (1985), which assumes that any past observed event is equally probable in the future for a given date. The main differences between our approach and ESP, including the implementation by Crochemore et al. (2017), lies in the lead times range: ours are less than one week, whereas ESP typically considers longer horizons. While this approach deserves further investigations, we consider it a no-skill baseline ensemble that represents the opposite extreme from the perfect rainfall forecast assumption.

**2.4. evaluation metric**

Metric choice: You use CRPS. Would CRPSS also be an option for comparison with the reference?

We thank the reviewer for this question. The Continuous Ranked Probability Skill Score (CRPSS) is indeed a valid option for relative performance assessment, using the climatology as reference. In this work, we focused on the absolute quality of the forecasts and therefore used the CRPS. Nevertheless, we will consider the CRPSS in the revised version to complement this analysis.

- L186: Please provide reference for this statement.

This sentence will be revised and completed with references as follows: "An ensemble forecast is considered reliable (or statistically consistent) when the ensemble spread adequately reflects forecasts uncertainty, such that the observations are statistically indistinguishable from the ensemble members (Buizza et al., 2005; Hamill, 2001; Talagrand et al., 1997; Whitaker & Loughe, 1998).

- I am not familiar with rank diagrams. How are they created? Why does the number of ranks differ between Figures 4 and 10?

We use the rank histogram to assess the reliability of the ensemble forecast with respect to a target. In Fig.4, it reflects the distribution of the evaluation period (2006-2008), as target, compared to the remain period (1989-2006) for the rainfall and the PET observations. In Fig.10, it depicts the distribution of the forecasted discharge through the forecasted ensemble. Regarding the difference in number of ranks between the two figures, there is no particular methodological reason, except a technical adaptation

related to the size of the ensembles. In Fig.4, each individual year (out of 18) was retained as a member, so the number of ranks matched the ensemble size. For Fig.10, because the members resulted from a combination of N seeds and M years, and for clearer visualization, the ensemble was binned into 10 classes using quantiles.

- L197: You can also provide a formula for the spread skill ratio

We will include the related formula in the revised version of the manuscript. The spread-skill ratio (SSR) (Whitaker & Loughe, 1998) provides insights into the consistency between the ensemble uncertainty and the actual forecast error. Specifically, given a time step $t$ (from a full evaluation period $T$), $y$ the observed target value, $x_i$ the forecast member $i$ (from the ensemble $N$), and $\bar{x}$ the mean of the ensemble, the SSR can be formulated as:

$$\text{SSR}$$

$$SSR = \frac{\sqrt{\frac{1}{T}\sum_{t=1}^{T}\sigma_t^2}}{\sqrt{\frac{1}{T}\sum_{t=1}^{T}(\bar{x}_t - y_t)^2}}$$

$$\text{Spread}$$

$$\sigma_t = \sqrt{\frac{1}{N-1}\sum_{i=1}^{N}(x_{t,i} - \bar{x}_t)^2}$$

Best values are expected to be close to 1, and indicate well-calibrated ensemble. Values below (or above) 1 indicate under-dispersion (or over-dispersion) of the ensemble.

The ranks (or classes) have been constructed using quantiles of the ensemble. Therefore, a more accurate label would be "ensemble quantile" or simply "rank".

2.5. Experimental settings

- L230: typos: '.', RR-> RQ? In Figure 5 you are using Rainfall - Discharge cross-correlation.

Thank you for these remarks, the "." is indeed a typographical error. Yes, the R-Q indicates Rainfall-Discharge cross-correlation. We used this information to approximate the input sequence length of the input features in the orchestrator.

- Figure 5: Please avoid overlapping colour areas, which will appear as a new colour. Perhaps just draw the lines.

Thank you for this suggestion. We will revise the figures, we will try with tiny lines highlight individual basins, which should avoid the overlapping issue.

- L239: Why 56 basins? Is the choice only motivated by NSE-values or also by different hydrological regimes?

The motivation behind this choice was primarily computational. Given the time and computation necessary to perform this approach, we did not find it necessary to run the ensemble climatology over all 531 basins. Regarding the selection approach, we used the NSE values, although other approaches such as hydrological regimes could also be considered.

Chapter 3 Results

- L340: Why should the observation be calibrated? Please provide reference.

By stating that the observation should be calibrated, we meant that the observations should be evenly distributed through the ensemble (Talagrand et al., 1997). However, this is more of an ideal expectation than a guaranteed outcome.

- fig 13: Are the low flows over- or underestimated?

As shown in Fig.10, there is no systematic bias for the low flows, except for the benchmark SACSMA (which exhibits overestimation) and to a lesser extent, the benchmark LSTM.

Chapter 4 Conclusions

- fig 14: see comment above

- fig 14: Why is there a spread on the left side for the deterministic optimal forecast? Is this due to the forecast initialization times and different lead times as a 'poor man's ensemble'??

We thank the reviewer for this insightful question. The spread observed on the left side for the deterministic optimal forecast originates from differences in model initializations (number of seeds) rather than from forcing uncertainty.  While this setup shares similarities with the "poor man's ensemble", commonly used in meteorological

forecasting, the forcing data were assumed to be perfectly known and identical across all forecasts and lead times. Thus, the spread reflects solely the effect of the model initialization variability.

- L397:" numerous publications": Please provide examples.

Thank you for this suggestion, we will provide examples of references to support this statement, including Kratzert et al. (2018, 2019), Feng et al. (2020, 2024) and Yang et al. (2025)

**References**

Buizza, R., Houtekamer, P. L., Pellerin, G., Toth, Z., Zhu, Y., & Wei, M. (2005). A Comparison of the ECMWF, MSC, and NCEP Global Ensemble Prediction Systems. *Monthly Weather Review*, *133*(5), 1076–1097. https://doi.org/https://doi.org/10.1175/MWR2905.1

Clark, M. P., Rupp, D. E., Woods, R. A., Zheng, X., Ibbitt, R. P., Slater, A. G., Schmidt, J., & Uddstrom, M. J. (2008). Hydrological data assimilation with the ensemble Kalman filter: Use of streamflow observations to update states in a distributed hydrological model. *Advances in Water Resources*, *31*(10), 1309–1324. https://doi.org/10.1016/j.advwatres.2008.06.005

Crochemore, L., Ramos, M.-H., Pappenberger, F., & Perrin, C. (2017). Seasonal streamflow forecasting by conditioning climatology with precipitation indices. *Hydrology and Earth System Sciences*, *21*(3), 1573–1591. https://doi.org/10.5194/hess-21-1573-2017

Day, G. N. (1985). Extended Streamflow Forecasting Using NWSRFS. *Journal of Water Resources Planning and Management*, *111*(2), 157–170. https://doi.org/10.1061/(ASCE)0733-9496(1985)111:2(157)

Feng, D., Beck, H., de Bruijn, J., Sahu, R. K., Satoh, Y., Wada, Y., Liu, J., Pan, M., Lawson, K., & Shen, C. (2024). Deep dive into hydrologic simulations at global scale: harnessing the power of deep learning and physics-informed differentiable models (δHBV-globe1.0-hydroDL). *Geoscientific Model Development*, *17*(18), 7181–7198. https://doi.org/10.5194/gmd-17-7181-2024

Feng, D., Fang, K., & Shen, C. (2020). Enhancing Streamflow Forecast and Extracting Insights Using Long-Short Term Memory Networks With Data Integration at Continental Scales. *Water Resources Research*, *56*(9), e2019WR026793. https://doi.org/https://doi.org/10.1029/2019WR026793

Hamill, T. M. (2001). Interpretation of Rank Histograms for Verifying Ensemble Forecasts. *Monthly Weather Review*, *129*(3), 550–560. https://doi.org/https://doi.org/10.1175/1520-0493(2001)129<0550:IORHFV>2.0.CO;2

Kratzert, F., Klotz, D., Brenner, C., Schulz, K., & Herrnegger, M. (2018). Rainfall–runoff modelling using Long Short-Term Memory (LSTM) networks. *Hydrology and Earth System Sciences*, *22*(11), 6005–6022. https://doi.org/10.5194/hess-22-6005-2018

Kratzert, F., Klotz, D., Shalev, G., Klambauer, G., Hochreiter, S., & Nearing, G. (2019). Towards learning universal, regional, and local hydrological behaviors via machine learning applied to large-sample datasets. *Hydrology and Earth System Sciences*, *23*(12), 5089–5110. https://doi.org/10.5194/hess-23-5089-2019

Talagrand, O., Vautard, R., & Strauss, B. (1997). Evaluation of probabilistic prediction systems. In *Workshop on Predictability, 20-22 October 1997*.

Whitaker, J. S., & Loughe, A. F. (1998). The Relationship between Ensemble Spread and Ensemble Mean Skill. *Monthly Weather Review*, *126*(12), 3292–3302. https://doi.org/https://doi.org/10.1175/1520-0493(1998)126<3292:TRBESA>2.0.CO;2

Yang, Y., Pan, M., Feng, D., Xiao, M., Dixon, T., Hartman, R., Shen, C., Song, Y., Sengupta, A., Delle Monache, L., & Ralph, F. M. (2025). Improving streamflow simulation through machine learning-powered data integration and its potential for forecasting in the Western U.S. *Hydrology and Earth System Sciences*, *29*(20), 5453–5476. https://doi.org/10.5194/hess-29-5453-2025